# A Spiking Transformer with Dynamic Threshold for Efficient Continual Learning

## Abstract

Although deep neural networks perform extremely well in controlled environments, they fail in real-world scenarios where the data isn't available all at once, and the model requires an update to adapt itself to the new data distribution, which might or might not follow the initial distribution. Previously acquired knowledge is lost during such subsequent updates from new data, a phenomenon commonly known as catastrophic forgetting. In contrast, the brain can learn without such catastrophic forgetting, irrespective of the number of tasks it encounters. Existing spiking neural networks (SNNs) for class-incremental learning (CIL) suffer a sharp performance drop as tasks accumulate. While Parameter-Efficient Fine-Tuning (PEFT) strategies have significantly mitigated this in non-spiking vision transformers by adapting minimal parameters but equivalent mechanisms in the spiking domain remain underexplored. We here introduce CATFormer (Context Adaptive Threshold Transformer), a scalable framework that overcomes this limitation. We observe that the key to preventing forgetting in SNNs lies not only in synaptic plasticity, but also in modulating neuronal excitability. At the core of CATFormer is the *Dynamic Threshold Leaky Integrate-and-Fire (DTLIF)* neuron model, which leverages context-adaptive thresholds as the primary mechanism for knowledge retention. This is paired with a Gated Dynamic Head Selection (G-DHS) mechanism for inference. Extensive evaluation on both static (CIFAR 10/100/Imagenet 100/Tiny-Imagenet 200) and neuromorphic (CIFAR10-DVS/SHD) datasets reveals that CATFormer outperforms existing rehearsal-free CIL algorithms across various task splits, establishing it as an ideal architecture for energy-efficient and class incremental learning.

## 1 Introduction

Progress in physical AI holds immense promise for enhancing real-world capabilities in avenues such as robotics, near-sensor edge devices, and autonomous systems. A critical challenge in these platforms is learning, as well as predicting cyclically over extended deployments with minimal resource utilisation. Model updates are often essential due to sequentially incoming data and the distributional shifts in such cases Chaudhry et al. (2019); Castro et al. (2018); Wang et al. (2024). But naively training standard deep neural networks (MLPs, CNNs, or even modern transformers) from scratch repeatedly typically results in *catastrophic forgetting* of previously acquired knowledge. In battery-operated, memory or bandwidth-constrained physical agents, data rehearsal (i.e., storage and replay of past data during training on new data) is often infeasible due to energy, onboard memory, privacy, or regulatory constraints Lesort et al. (2020).

Energy-efficient learning architectures are essential in these applications, and **Spiking neural networks (SNNs)** have become a well-established solution, offering event-driven sparse computations. Recent advances have brought class-incremental learning (CIL) to SNNs with early efforts mainly on small-scale tasks (e.g., CIFAR-10) and, more recently, on larger, more realistic benchmarks like CIFAR-100 using various incremental task regimes Ni et al. (2025); Han et al. (2023). However, prior efforts in SNN-based CL have predominantly relied on convolutional (CNN) architectures. Recently, transformers and their variants have dominated performance in numerous AI applications. Vision transformers Dosovitskiy et al. (2021) can also capture global dependencies along with contextual understandings. SpikFormer Zhou et al. (2023b), QKformer Zhou et al. (2024) and Spiking-Former Zhou et al. (2023a) extends the standard vision transformer paradigm to SNNs by combining

its strengths with energy efficiency. In case of pretrained ViTs there have been PEFT methods for continual learning, such as Wang et al. (2022b),Liang & Li (2024); Gidaris & Komodakis (2018). Although these methods work significantly well, the transferability of these methods to the spiking domain remains highly unexplored. Hence, we design CATFormer here, a general framework for SNN-based transformers in class incremental learning.

CATFormer is inspired by the brain, where resistance to forgetting has been proposed to be closely linked to neuromodulation Masse et al. (2018); Beaulieu et al. (2020). Neuromodulators such as acetylcholine, dopamine, serotonin, and norepinephrine mediate changes in neural circuit behaviour via alterations in plasticity or excitability. For instance, acetylcholine plays a central role by modulating membrane excitability and synaptic plasticity in hippocampal and cortical networks, thereby enabling the rapid encoding of new memories while transiently lowering neuronal firing thresholds to suppress interference from prior information Hasselmo (1999); Grossberg (2017). Similar cholinergic modulation of plasticity is also observed in rodent piriform cortex Hasselmo & Barkai (1995). These processes in the brain dynamically regulate neuronal excitability and firing thresholds Liu et al. (2022); Oh & Disterhoft (2015), enabling selective pathway activation and memory consolidation while preventing interference Xu et al. (2005); Farmer & Thompson (2012). Computational models and experimental studies further demonstrate that neuromodulators broadly demonstrate task and context-specific routing of information by reshaping network dynamics Tsuda et al. (2021); Masse et al. (2018); Hammouamri et al. (2022). Although we don't claim CATFormer to be completely biologically plausible, these multi-scale neuromodulatory effects inspire our approach to implement adaptive thresholds within CATFormer, serving as an analogy to support plasticity in exemplar-free continual learning.

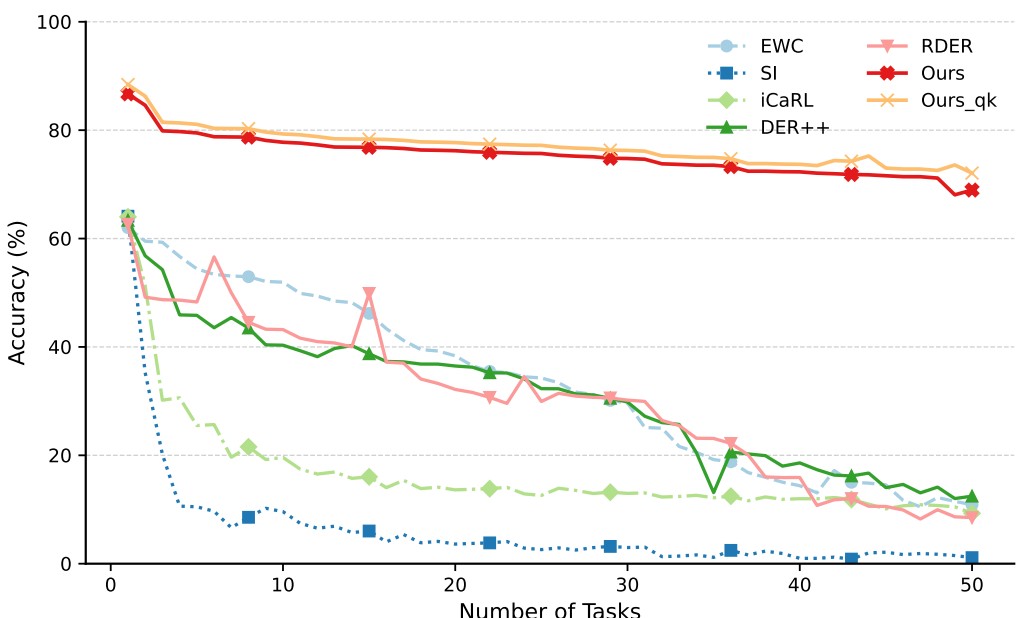

Figure 1: Test performance variation with respect to the progress in the number of trained tasks (for a maximum of 50 tasks). CATFormer (ours) maintains a consistent performance compared to the other existing CIL methods when implemented on a Spiking Transformer. All methods are evaluated on CIFAR 100.

In this work, we systematically study class incremental learning in a spiking vision transformer such as SpikFormer and QKFormer Zhou et al. (2023b; 2024). We analyse how biologically inspired, dynamic spiking thresholds influence continual learning and propose mechanisms that foster robustness regardless of the number of encountered tasks. This is crucial for *physical AI and robotics*: real-world robots and edge agents must adapt over months or years, often encountering dozens or even hundreds of different skills or operating regimes Lesort et al. (2020). Thus, we rigorously

evaluate our approach at unprecedented scales, including challenging 50 and 100-task sequences, providing a rigorous testbed for long-term continual learning relevant to autonomous robotics and physical edge applications. Moreover, our data rehearsal-free protocol ensures that results directly reflect core algorithmic advances, rather than storage-based workarounds. This can be observed in Figure 1, where the model maintains its accuracy across longer task sequences. We observe a phenomenon we call *reverse forgetting*, where the model actually learns more effectively when exposed to fewer classes per task. This scenario is more reflective of real-world settings for example, in robotics or lifelong learning, the model is unlikely to encounter 20 or 50 new classes all at once Lesort et al. (2020); Masse et al. (2018).

**We make the following advances in this work** [1] **:**

- To the best of our knowledge, we present the first CIL framework designed for spiking vision transformers, closing a major gap in the field.

- We propose a novel, biologically-inspired adaptation mechanism where a frozen backbone learns new tasks primarily through task-specific dynamic thresholds. This approach serves as the core mechanism for preventing catastrophic forgetting without requiring network growth or the storage of raw data exemplars.

- We demonstrate state-of-the-art performance among exemplar-free SNNs and, critically, show that CATFormer is uniquely scalable to long task sequences. Unlike prior methods that degrade, our model maintains or even improves its accuracy on challenging benchmarks of up to 100 incremental tasks, i.e., depicting a **reverse forgetting** trend.

- We introduce CATFormer effectively as a PEFT method tailored for the spiking domain. However, instead of introducing external adapters, we leverage the modulation of intrinsic neuronal thresholds to adapt the spiking transformer to new tasks.

- The architecture is at least 8.6 times energy efficient compared to ViTs for class incremental learning on pretrained architecture, and on non-pretrained it is at least 21.4 times more energy efficient (Table 10).

## 2 RELATED WORK

### 2.1 CONTINUAL LEARNING PARADIGMS

Continual learning methods often fall into three main approaches Van de Ven & Tolias (2019). **Regularisation-based** methods mitigate forgetting by constraining updates to parameters deemed important for previous tasks, thus preventing overwriting of past knowledge Kirkpatrick et al. (2017); Zenke et al. (2017). **Rehearsal-based** methods maintain a buffer of stored previous data samples either as original images or as representations for replay during training, improving memory retention but incurring increased storage and privacy concerns Rebuffi et al. (2017); Buzzega et al. (2020); Zhu et al. (2022); Ye & Bors (2025). **Architecture-based** methods adapt the network structure dynamically Han et al. (2023); Wang et al. (2025), such as by adding modules or selecting task-specific sub-networks, balancing plasticity and stability at the cost of computational overhead Rusu et al. (2016); Fernando et al. (2017). While these architecture-based continual learning methods are effective, they include limitations of scalability and memory overhead as the task count increases.

In the brain, context-dependent signals from regions like the prefrontal cortex project across cortical areas, allowing neural circuits to adaptively process information based on the task at hand Engel et al. (2001); Johnston et al. (2007); Miller & Cohen (2001). Previous techniques leveraged EWC Kirkpatrick et al. (2017) with a gating mechanism to stabilise training for feedforward and recurrent architectures Masse et al. (2018).

### 2.2 TRANSFORMERS IN CONTINUAL LEARNING

Transformers, with their self-attention mechanisms, have recently emerged as a powerful alternative to convolutional neural networks (CNNs) for continual learning due to their capability to model

---

[1]We will make the code public upon acceptance of the paper.

long-range dependencies and easy extension of pretrained architectures Wang et al. (2025); Liang & Li (2024). Recent work has focused on parameter-efficient fine-tuning techniques, such as Low-Rank Adaptation (LoRA), to continually update large pre-trained transformers with minimal overhead. Prominent among these are prompt based methods such asWang et al. (2022a;b); Smith et al. (2023), adapter based such as Chen et al. (2022) and LoRA based methods such as Liang & Li (2024); Wu et al. (2025). However, these standard vision transformers require computations of attention, which in turn leads to energy inefficiency due to their heavy matrix multiplications. Hence, we move towards spiking vision transformers (**atleast 3.31 times energy efficient**) Zhou et al. (2023b; 2024), which can get rid of these heavy computations. Research on continual learning for spiking vision transformers remains largely unexplored. Our work presents data rehearsal-free continual learning on spiking vision transformers trained both from scratch and pretrained, addressing these challenges and expanding the landscape of SNN continual learning.

### 2.3 Spiking Neural Networks for Continual Learning and Neuromodulation Inspiration

Early SNN continual learning methods adapted classical regularisation and rehearsal methods that are limited to CNNs Han et al. (2023); Lin et al. (2025). Recent work Ni et al. (2025) shows significant improvement by incorporating rehearsal buffers into a method that was inspired by DER++ Buzzega et al. (2020) to enhance performance, but violates data privacy and is memory inefficient. A preliminary work in converting the idea of neuromodulation to circuit level was demonstrated by Hammouamri et al. (2022), where the thresholds of the current layer were modulated based on the previous layer's excitation. Our model introduces adaptive, task-specific dynamic thresholds. This mimics neuromodulatory circuits that release context signals modulating downstream neurons' responses, corresponding to our task-ID routing mechanism that dynamically tunes neuron thresholds per task to enable data replay-free continual adaptation.

### 2.4 Dynamic Thresholds in Spiking Neural Networks

Adaptive firing thresholds have been shown to improve temporal precision and robustness in SNNs Ding et al. (2022); Huang et al. (2016); Wei et al. (2023). However, prior work has predominantly focused on single-task adaptation rather than task-specific thresholding for continual learning. By freezing synaptic weights after the initial task and relying exclusively on learnable, per-task thresholds for plasticity, our approach introduces a novel mechanism for preventing catastrophic forgetting in a rehearsal-free framework. Together with lightweight task gating, this enables stable continual learning across up to 100 tasks on static and neuromorphic datasets, setting a new standard in memory and energy-efficient SNN continual learning.

## 3 Methodology

We introduce a data rehearsal-free framework for class-incremental learning (CIL) in Spiking Neural Networks (SNNs) that robustly accommodates new classes over time without catastrophic forgetting. Unlike existing systems, our entire system is trained without the use of a separate exemplar representation buffer from previous learning stored across the tasks Ni et al. (2025); Buzzega et al. (2020). The proposed architecture leverages two key innovations: *task-specific (Context Adaptive) dynamic neuronal thresholds* and a *gated inference mechanism*, combined through a two-stage training protocol.

### 3.1 Problem Formulation and Notation

In CIL, the model is exposed to $T$ number of tasks $\{\mathcal{T}_0, \mathcal{T}_1, \ldots, \mathcal{T}_{T-1}\}$ sequentially, each with dataset $\mathcal{D}^k = \{(x_i^k, y_i^k)\}_{i=1}^{N_k}$ and disjoint label sets $\mathcal{Y}_k$ (i.e., $\mathcal{Y}_i \cap \mathcal{Y}_j = \emptyset$ for $i \neq j$). At each task $k$, the model must classify samples over the cumulative label space $\mathcal{Y}_{1:k}$, having access solely to the current task's train data during, and *without the any task oracle or previous task samples* at test time.

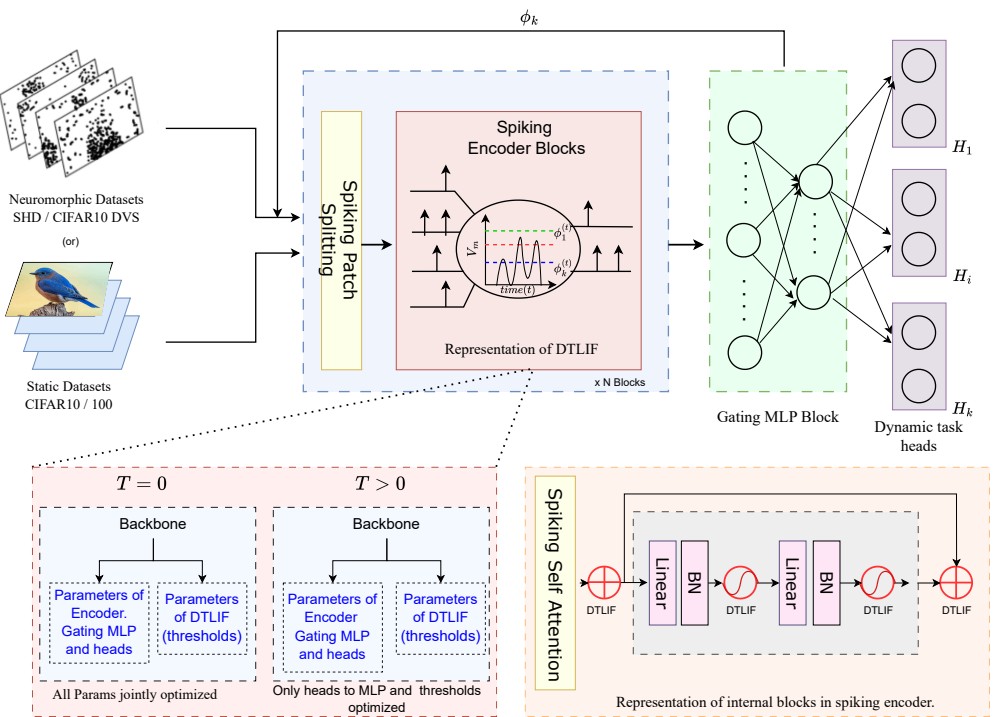

Figure 2: The diagram depicts the full architecture and workflow of CATFormer.

## 3.2 CONTEXT ADAPTIVE DYNAMIC THRESHOLD NEURONS

**Dynamic Threshold LIF Neuron Model:** To enable task-adaptive spiking dynamics, we extend the standard Leaky Integrate-and-Fire (LIF) neuron with *context adaptive, learnable thresholds*. Updation of thresholds : $\tilde{V}_j^{(t)} = \left(1 - \frac{1}{\tau}\right)V_j^{(t-1)} + \frac{1}{\tau}I_j^{(t)}$ where $\tau$ is the membrane time constant, $I_j^{(t)}$ is the input current. $S_j^{(t)} = \Theta\big(\tilde{V}_j^{(t)} - \phi_j^{(k)}\big)$ here $\Theta(\cdot)$ is the Heaviside step function and $S_j^{(t)}$ is the spike output. We use soft reset Mechanism $V_j^{(t)} = \tilde{V}_j^{(t)} - S_j^{(t)}\,\phi_j^{(k)}$. The implementation of this can be referred to in the Appendix 1.

**Updation of Dynamic thresholds:** During training on task $k$, the threshold $\phi_c^{(k)}$ is updated via gradient descent: $\phi_j^{(k)} \leftarrow \phi_j^{(k)} - \eta\frac{\partial\mathcal{L}}{\partial\phi_j^{(k)}}$ where $\eta$ is the learning rate and $\mathcal{L}$ is the loss function. This mechanism allows each channel to adjust its firing threshold for different tasks, supporting task-adaptive spiking in continual learning.

## 3.3 TWO-STAGE TRAINING PROTOCOL

Our training protocol balances plasticity and stability through a two-stage approach: joint optimization for the initial task, followed by selective parameter freezing for subsequent tasks. This design prevents catastrophic forgetting while maintaining the ability to learn new classes. Algorithm 1 outlines the complete procedure.

**Stage 1: Initial Task Learning** ($k = 0$). For the first task $\mathcal{T}_0$, we jointly optimize the backbone parameters $\theta$, the initial thresholds $\phi^{(0)}$, and the first classification head $W_0$ using cross-entropy loss on dataset $\mathcal{D}^0$. This phase establishes the feature extraction capabilities of the network. After training completes, the backbone $\theta$ is frozen and will not be updated for any subsequent task.

**Stage 2: Incremental Learning** ($k > 0$). When a new task $\mathcal{T}_k$ arrives, we freeze all previously learned components: the backbone $\theta$, all previous threshold configurations $\{\phi^{(0)}, \ldots, \phi^{(k-1)}\}$, and all previous classification heads $\{W_0, \ldots, W_{k-1}\}$. We then initialize a new classification head $W_k$ (using Xavier initialization) and a new threshold set $\phi^{(k)}$ (initialized to $\phi_{init} = 0.5$). Only these new parameters are optimized on the current task's data:

$$\min_{\{W_k, \phi^{(k)}\}} \mathbb{E}_{(x,y) \sim \mathcal{D}^k} \left[ \mathcal{L}_{CE}(W_k \cdot f(x; \theta, \phi^{(k)}), y) \right] \tag{1}$$

This selective optimization ensures that (i) old task performance is preserved since frozen parameters cannot be degraded, and (ii) new task learning is unrestricted since new parameters are fully trainable.

---

**Algorithm 1** CATFormer Training Protocol

---

1: **Input:** Task sequence $\{\mathcal{T}_k, \mathcal{D}^k\}_{k=0}^{T-1}$
2: **Initialize:** Backbone $\theta$, base threshold $\phi_{init} = 0.5$, gating MLP $\mathcal{G}$, gating dataset $\mathcal{G}_{train} = \emptyset$
3: **for** $k = 0$ to $T - 1$ **do**
4:     Add head $W_k$ (Xavier init), set $\phi^{(k)} \leftarrow \phi_{init}$
5:     **if** $k = 0$ **then**
6:         Train $\{\theta, \phi^{(0)}, W_0\}$ jointly with $\mathcal{L}_{CE}$ on $\mathcal{D}^0$
7:         Freeze backbone $\theta$ permanently
8:     **else**
9:         Freeze: $\theta, \{\phi^{(0)}, \ldots, \phi^{(k-1)}\}, \{W_0, \ldots, W_{k-1}\}$
10:         Jointly optimize $\{W_k, \phi^{(k)}\}$ on $\mathcal{D}^k$ with $\mathcal{L}_{CE}$
11:     **end if**
12:     {Build gating dataset and update gating network}
13:     **for** each sample $(x, y) \in \mathcal{D}^k$ **do**
14:         Extract features using base thresholds: $\mathbf{f}(x) \leftarrow \text{SpikFormer}(x; \theta, \phi_{init})$
15:         Add to gating dataset: $\mathcal{G}_{train} \leftarrow \mathcal{G}_{train} \cup \{(\mathbf{f}(x), k)\}$
16:     **end for**
17:     Train gating MLP $\mathcal{G}$ on $\mathcal{G}_{train}$ with $\mathcal{L}_{CE}$
18: **end for**

---

**Gating Network Construction and Training.** After completing training on each task $k$, we need to update the gating network $\mathcal{G}$ to recognize the new task. We construct the gating dataset $\mathcal{G}_{train}$ by extracting features from all samples in the current task's dataset $\mathcal{D}^k$ using the base thresholds $\phi_{init}$. These feature-label pairs $(\mathbf{f}(x), k)$ are accumulated across all tasks seen so far, creating a growing dataset that represents the feature distributions of all tasks.

Crucially, we store only the extracted features rather than the raw input images. Since features are extracted once per task using the fixed $\phi_{init}$, this provides a compact representation with memory overhead bounded by the feature dimension $D$ rather than the input image dimensions. This is fundamentally different from exemplar replay methods that must store samples from all previous tasks and manage buffer sizes.

The gating network i.e the head is trained $k > 0$on the accumulated $\mathcal{G}_{train}$ using standard cross-entropy loss to predict task identity from features. We use a lower learning rate $\eta_g = 0.1 \times \eta_\phi$ when training the gating network to promote stability and reduce overfitting to recently added tasks. We ensure the weights are normalized. This ensures balanced task prediction accuracy across all seen tasks.

## 3.4 INFERENCE VIA DYNAMIC HEAD ROUTING

During inference, we employ our Gated Dynamic Head Selection (G-DHS) mechanism to efficiently route inputs to appropriate task-specific heads without requiring a task oracle.

**Gating MLP Architecture.** The gating network is a two-layer MLP that maps feature embeddings to task predictions: $\mathcal{G}(\mathbf{f}) = \text{Linear}(\text{ReLU}(\text{Linear}(\mathbf{f})))$, where $\mathbf{f} \in \mathbb{R}^D \rightarrow \mathbb{R}^{D/4} \rightarrow \mathbb{R}^k$. Given input $x$ where $D$ is the feature dimension, our inference procedure follows A.1

| Backbone | Methods | Number of Tasks | | | | Parameters (M) | |
|---|---|---|---|---|---|---|---|
| **Traditional** | | **1 Task (Full dataset)** | | | | **Total** | |
| VGG-11 | Hybrid | 67.87 | | | | 9.27 | |
| ResNet-19 | TET | 74.47 | | | | 12.63 | |
| SpikFormer-4-384 | BPTT | 77.86 | | | | 9.32 | |
| **Class incremental Learning** | | **10 Tasks** | **20 Tasks** | **25 Tasks** | **50 Tasks** | **Task 0** | **Task k** |
| VGG-8 | DSD-SNN | $60.47 \pm 0.72$ | $57.39 \pm 1.97$ | $53.79 \pm 2.67$ | $50.55 \pm 1.76$ | 14.2 | 14.2 |
| SpikFormer-4-384 | EWC | $18.81 \pm 1.22$ | $17.06 \pm 0.83$ | $15.73 \pm 0.38$ | $9.73 \pm 0.62$ | 9.32 | 18.64 |
| | SI | $7.98 \pm 0.33$ | $4.66 \pm 0.74$ | $3.22 \pm 0.14$ | $1.84 \pm 0.09$ | 9.32 | 27.96 |
| | iCARL | $33.46 \pm 1.52$ | $28.42 \pm 0.77$ | $22.37 \pm 0.90$ | $10.89 \pm 1.19$ | 9.32 | 9.32 |
| | DER++ | $34.99 \pm 1.39$ | $28.48 \pm 1.16$ | $24.9 \pm 1.07$ | $13.12 \pm 3.2$ | 9.32 | 9.32 |
| | RDER | $15.82 \pm 0.36$ | $14.65 \pm 0.27$ | $12.28 \pm 0.59$ | $8.8 \pm 0.47$ | 11.06 | 11.06 |
| | **Ours** | $\mathbf{68.33 \pm 4.51}$ | $\mathbf{69.13 \pm 2.36}$ | $\mathbf{71.34 \pm 1.75}$ | $\mathbf{75.66 \pm 2.72}$ | 10.5 | **0.23** |

Table 1: Comparison of Average Accuracy (AA%) on Split CIFAR-100 across different task granularities. DSD-SNN results for 25/50 tasks from Han et al. (2023); other CIL baselines Kirkpatrick et al. (2017); Zenke et al. (2017); Rebuffi et al. (2017); Buzzega et al. (2020); Yan et al. (2021) evaluated on SpikFormer Zhou et al. (2023b). Task 0 and Task k columns show parameter counts (in millions) for initial and subsequent task training.

# 4 RESULTS

## 4.1 DATASET AND EXPERIMENTAL SETUP

**Evaluation Metrics.** We evaluate CATFormer across diverse static and neuromorphic datasets to demonstrate the effectiveness of our context-adaptive dynamic threshold mechanism. Our benchmark suite includes:

**Static datasets:** CIFAR-10/100 Krizhevsky et al. (2009) (10/100 classes, $32 \times 32$ RGB images), Tiny-ImageNet Le & Yang (2015) (200 classes, $64 \times 64$ images), and ImageNet100 (100 classes, $224 \times 224$ resolution).

**Neuromorphic datasets:** CIFAR10-DVS Li et al. (2017) (10 classes, event-based DVS recordings) and SHD Cramer et al. (2020) (20 classes, neuromorphic auditory spike sequences).

## 4.2 CLASS INCREMENTAL LEARNING ON STATIC DATASETS

### 4.2.1 PERFORMANCE ON CIFAR-100

**Backbone Architecture Selection.** We evaluate CATFormer using two spiking transformer architectures: SpikFormer Zhou et al. (2023b) and QKFormer Zhou et al. (2024). SpikFormer serves as our primary backbone for most experiments. QKFormer, featuring hierarchical attention and improved parameter efficiency, is employed for pretrained model evaluations (comparing against PEFT methods) and extended benchmarks (Permuted MNIST, ImageNet100). This dual-backbone evaluation demonstrates that dynamic threshold modulation is architecture-agnostic.

We compare CATFormer against state-of-the-art class incremental learning (CIL) methods on Split CIFAR-100. All methods use the either SpikFormer or QKFormer as backbone. Table 1shows the existing methods suffer from catastrophic forgetting, CATFormer design exhibits a stable performance across task splits.

**Task 0 Parameter Allocation.** Table 1 shows that Task 0 trains 10.5M parameters compared to 9.32M for the baseline SpikFormer. This includes: (i) the backbone $\theta$ (9.32M), (ii) initial thresholds $\phi^0$ (0.20M), (iii) the first head $W_0$ (0.03M), and (iv) the gating network (1.1M). Joint optimisation establishes strong initial representations. For tasks $k > 0$, only 0.23M parameters (thresholds and heads) are trained, demonstrating efficiency through threshold modulation alone.

Classical regularization methods (EWC Kirkpatrick et al. (2017), SI Zenke et al. (2017)) exhibit severe forgetting, with accuracy collapsing to below 10% at 50 tasks. Rehearsal-based approaches (iCaRL Rebuffi et al. (2017), DER++ Buzzega et al. (2020)) perform better but remain constrained by memory requirements particularly challenging for neuromorphic hardware like Loihi 2 Shrestha

et al. (2024), where even the 2000-sample buffer proposed by ALADE-SNN Ni et al. (2025) is difficult to accommodate.

Among rehearsal-free SNN methods, DSD-SNN Han et al. (2023) represents the previous state-of-the-art, achieving 60.47% on 10 tasks. However, it follows the expected forgetting pattern: accuracy degrades consistently to 50.55% at 50 tasks. We extended the original DSD-SNN repository[2] to verify this trend across 25 and 50 tasks.

CATFormer fundamentally reverses this pattern. Rather than degrading with more tasks, our model improves from 68.33% (10 tasks) to 75.66% (50 tasks) a counter-intuitive "reverse forgetting" phenomenon illustrated in Figure 3( in appendix A.4.) This behavior stems from our dynamic threshold adaptation with gating mechanism, which optimizes neuronal firing for new tasks while preserving prior knowledge. The progressive improvement suggests that lesser the number of classes the model provide richer context for threshold optimization, enabling better feature discrimination across the entire task sequence.

This reverse forgetting trend is particularly significant for real-world continual learning scenarios in robotics and embodied AI Lesort et al. (2020); Hajizada et al. (2022), where systems must adapt to continuous data drift and evolving task distributions over extended deployment periods.

**Parameter Efficiency Analysis**   Beyond accuracy, CATFormer demonstrates exceptional parameter efficiency. The base SpikFormer architecture contains 9.32M parameters, while our model has 9.3M parameters (1.1M of which are attributed to the routing mechanism). At task $k$, only 0.23M parameter updates are performed (Table 1).

The efficiency advantage becomes significant during continual learning. At task $T_0$, CATFormer trains 10.5M parameters similar to baseline methods. However, for subsequent tasks ($T_k$ where $k > 0$), we update only 0.23M parameters while baselines retrain their entire models and often maintain memory buffers, which introduces privacy concerns.

The memory footprint for task-specific thresholds is minimal: approximately 16,032 thresholds per task require only 64.2 KB when stored as floating-point 32-bit (FP32) values. This can be further reduced to 32.1 KB using FP16 precision without performance degradation, as threshold precision requirements are modest. Thus, our model scales efficiently with $\mathcal{O}(T)$ memory complexity, where $T$ is the number of tasks.

### 4.2.2   EXTENDED EVALUATION: CIFAR-10

On the 10-class CIFAR-10 dataset, CATFormer achieves 89.29% accuracy for the 5-task split (Table 5), substantially outperforming the previous best rehearsal-free method, DSD-SNN (80.39%). Additional CIFAR-10 evaluations are provided in Appendix A.7.

### 4.3   COMPARISON WITH CIL METHODS FOR PRE-TRAINED MODELS

Recent advances in continual learning for vision transformers leverage parameter-efficient fine-tuning (PEFT) techniques on a pre-trained backbone model, such as prompt tuning (L2P Wang et al. (2022b), DualPrompt Wang et al. (2022a), CODA-Prompt Smith et al. (2023)), low-rank adaptation (LoRA variants) Gidaris & Komodakis (2018); Liang & Li (2024), and adapter-based methods. These approaches introduce small task-specific parameter sets while freezing the pretrained backbone, enabling efficient adaptation with minimal overhead.

**Our approach differs fundamentally: instead of auxiliary prompts or adapters, we modulate neuron membrane thresholds**. This is a lightweight, biologically inspired mechanism that requires no architectural modifications. This strategy maintains expressive power while dramatically reducing trainable parameters.

Table 2 reveals several insights. First, while PEFT methods like SD-LoRA achieve strong 10-task performance (88.01%), they struggle to scale: SD-LoRA runs out of memory at 20 tasks despite having fewer trainable parameters than CATFormer. Second, among SNN-based methods, CATFormer substantially outperforms DSD-SNN, while using 10 times fewer trainable parameters per

---

[2]https://github.com/BrainCog-X/Brain-Cog.git

| Methods | 10 tasks | 20 tasks | 50 tasks | Base param | param/task k |
|---|---|---|---|---|---|
| L2P | 83.18±1.20 | 79.51±0.67 | 67.95±2.12 | 172 | 0.12 |
| DualPrompt | 81.48±0.86 | 80.44±1.38 | 72.5± 1.08 | 172 | 0.86 |
| CODA-Prompt | 86.31±0.12 | 81.36±0.88 | 73.77 ± 0.98 | 172 | 4.6 |
| InfLoRA | 86.75±0.35 | 80.97±0.74 | 70.68 ± 1.26 | 172 | 0.51 |
| SD-LoRA | 88.01±0.31 | OOM | OOM | 172 | 0.39 |
| QKFormer (ours) | 71.92±0.85 | 72.15 ± 1.08 | **77.89**±**1.45** | **17.4** | **0.48** |

Table 2: CIFAR-100 class-incremental learning comparison between pretrained ViT with PEFT methods v/s pretrained QKFormer on our method for different task granularity. Here the Parameters are in millions

task (0.23M vs. 14.2M). Third, our parameter efficiency enables scaling to much longer task sequences, as demonstrated in Table 1.

To examine PEFT transferability to spiking architectures, we applied various techniques to QK-Former (Table 6). Standard LoRA performs poorly (60.48%), and even spiking-adapted variants underperform our threshold modulation approach (71.60%). Notably, replacing attention with identity mappings yields only 18.50% accuracy, confirming that structured feature extraction is essential. Our dynamic threshold approach achieves strong performance (71.60%) without additional architectural components. Detailed section is provided in appendixA.8

### 4.4 EVALUATION ON IMAGENET100 AND EXTENDED EVALUATION: CIFAR-10

At 50 tasks on Imagenet100 3, the reverse forgetting effect becomes pronounced: both Spik-Former and QKFormer with dynamic thresholds improve substantially from their 10-task performance (75.66% and 77.89% respectively), while L2P degrades from 83.18% to 67.95%. QKFormer achieves the highest accuracy among all methods, demonstrating that threshold modulation scales effectively with long task sequences. Table can be referred from appendix 4.5.

### 4.5 IMAGENET100

| Methods | SNN | CIFAR-100 | | ImageNet-100 | |
|---|---|---|---|---|---|
| | | 10 tasks | 50 tasks | 10 tasks | 50 tasks |
| L2P | No | 83.18±1.20 | 67.95±2.12 | 86.12 ± 1.01 | 69.49 ± 0.55 |
| SD-LoRA | No | 88.01±0.31 | OOM | 82.13 ± 0.88 | OOM |
| QKFormer + Ours | Yes | 71.92±0.85 | **77.89**±**1.45** | 70.56 ± 1.44 | **71.3 ± 0.95** |

Table 3: Comparison of CIL performance on CIFAR-100 (10/50 tasks) and ImageNet-100 (10/20 tasks).

On the 10-class CIFAR-10 dataset, CATFormer achieves 89.29% accuracy for the 5-task split (Table 5), substantially outperforming the previous best rehearsal-free method, DSD-SNN (80.39%). Additional CIFAR-10 evaluations are provided in Appendix A.7.

### 4.6 MEMORY STABILITY AND LEARNING PLASTICITY

CATFormer exhibits consistently low BWT across all evaluated settings. This transfer demonstrates that dynamic thresholds enable the network to maintain performance on previous tasks while learning new ones. The detailed section is provided in the appendix A.9

### 4.7 ABLATION STUDIES

We conducted targeted ablations on Split CIFAR-10 (5 tasks, 2 classes each) to isolate the contribution of each component. Results in Table 4 reveal several critical insights.

**Fixed Threshold:** Using static learnable threshold, task-invariant thresholds causes severe catastrophic forgetting (42.87± 1.26% average). While first-task performance is reasonable (72.59%),

| Ablation Variant | Avg. Accuracy | Task 0 Accuracy |
|---|---|---|
| Fixed learnable Threshold | $42.87 \pm 1.26$ | $72.59 \pm 1.86$ |
| SpikIdentityFormer | $59.38 \pm 0.98$ | $70.62 \pm 1.75$ |
| Random Identity Former | $53.17 \pm 2.13$ | $62.43 \pm 0.99$ |
| FFN Frozen | $63.24 \pm 1.78$ | $72.17 \pm 1.59$ |
| **CATFormer (Full)** | $\mathbf{89.29 \pm 2.53}$ | $\mathbf{93.87 \pm 0.45}$ |

Table 4: Ablation study on CIFAR-10 (5 tasks). Average accuracy across all tasks and first-task accuracy are reported.

each subsequent task triggers 15-18% degradation the classic forgetting signature. This highlights that adaptive thresholds are essential for continual learning in SNNs, a factor overlooked by prior work Ni et al. (2025); Han et al. (2023); Shen et al. (2024). The biological substrate for such threshold dynamics may involve neuromodulation Tsuda et al. (2021); Oh & Disterhoft (2015); Liu et al. (2022); Xu et al. (2005).

**SpikIdentityFormer:** Replacing all attention modules with identity mappings (following MetaFormer observations Yu et al. (2022; 2024)) yields 59.38% accuracy substantially below full CATFormer but above fixed thresholds. This indicates that while structured token mixing is beneficial, threshold adaptation provides the primary continual learning capability.

**Random Identity Former:** Replacing attention with random uniform values degrades performance further to 53.17%, confirming that structured feature transformation matters. The 6.21% gap between random and identity variants demonstrates that disrupting information flow is more harmful than simply removing it.

**FFN Frozen:** Freezing feed-forward networks while allowing attention to adapt yields 63.24% accuracy. Comparing with SpikIdentityFormer (frozen attention, learnable FFN at 59.38%), we see that learnable attention provides a 3.86% advantage, indicating both components contribute to continual learning.

The full CATFormer (89.29%) substantially outperforms all ablations, confirming that both architectural components (structured feature extraction via attention and feed-forward layers) and the core innovation (dynamic threshold modulation) are essential for effective rehearsal-free continual learning.

## 5 DISCUSSION

CATFormer shows that biologically inspired dynamic threshold adaptation enables rehearsal-free continual learning in SNNs, maintaining or improving accuracy across up to 50 tasks. Its competitive performance, reverse forgetting, and low memory cost make it well suited for resource-constrained embodied systems. Unlike rehearsal-based methods, which incur prohibitive storage, energy, and privacy costs in robotics Lesort et al. (2020); Hajizada et al. (2022), CATFormer removes replay buffers via task-specific neuronal plasticity. This is crucial since storing even 2,000 CIFAR-100 samples requires 25–30 MB Ni et al. (2025), and methods relying on network expansion or pruning create unstable resource demands for embedded or neuromorphic platforms.Dynamic thresholds provide a compact, computation-efficient solution: CATFormer requires only 0.23M trainable parameters and 64 KB of threshold storage per task, with minimal routing overhead (1.1M parameters vs. the 9.32M-parameter SNN backbone). This biologically grounded, hardware-efficient design matches the needs of autonomous systems operating over long lifetimes. Future work should explore threshold adaptation under streaming, non-stationary conditions and investigate neuromorphic deployment on Loihi 2 Shrestha et al. (2024).

## 6 REPRODUCIBILITY STATEMENT

We provide dataloaders details, hyperparameter settings, hardware and software configurations, and code snippet of our DT-LIF implementation class (Appendix A.2, A.11, A.12, A.14). We commit to

sharing our codebase upon acceptance of the paper. We have thoroughly checked the implementation and ensured presentation of statistically sound empirical results.

## 6.1 ETHICS STATMENT

Continual learning involves repeated training of neural networks. This requires significant energy. Spiking neural networks offer a multiplication-free, extremely low-energy computation solution. Hence, they are environmentally friendly and can potentially prolong battery life. Moreover, although data rehearsal and continual learning approaches perform well, storage of representative training samples can lead to potential privacy infringement. Hence, we believe research on energy-efficient data rehearsal-free approaches, such as a CATFormer, is of paramount importance for green and privacy-preserving deep learning. Having said this, a potential pitfall of enabling continual learning in deep neural networks, such as a transformer, is that without any additional regulations in CATFormer, models can update their behaviour on non-designated train data samples, thereby still violating privacy and causing spurious predictions.

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

# A  APPENDIX

## A.1  EXTENDED METHODS

:

---

**Algorithm 2** Gated Inference

---

1: **Input:** Test sample $x$, trained model $\{\theta, \Phi, \{W_0 \cdots W_k\}, \mathcal{G}\}$, number of seen tasks $k$
2: **Output:** Predicted class $\hat{y}$
3: **if** $k = 0$ **then**
4:    Set thresholds to $\phi^{(0)}$ and reset SNN state
5:    Extract features: $\mathbf{f}(x) \leftarrow$ SpikFormer or QKFormer$(x; \theta, \phi^{(0)})$
6:    **return** $\arg\max(W_0 \mathbf{f}(x))$
7: **else**
8:    **Task Prediction:**
9:       Set all thresholds to base $\phi_{init}$ and reset SNN state
10:      Extract base features: $\mathbf{f}_{\text{base}}(x) \leftarrow$ SpikFormer or QKFormer$(x; \theta, \phi_{init})$
11:      Predict task: $k^* \leftarrow \arg\max(\mathcal{G}(\mathbf{f}_{\text{base}}(x)))$
12:   **Classification (once the head is selected):**
13:      Set thresholds to $\phi^{(k^*)}$ and reset SNN state
14:      Extract task-specific features: $\mathbf{f}_{k^*}(x) \leftarrow$ SpikFormer or QKFormer$(x; \theta, \phi^{(k^*)})$
15:      **return** $\arg\max(W_{k^*} \mathbf{f}_{k^*}(x))$
16: **end if**

---

The inference process operates in two stages. First, we use base thresholds $\phi_{init}$ to extract features in a consistent space across all tasks, allowing the gating network to reliably predict task identity. Once the task is identified, we reconfigure the network with task-specific thresholds $\phi^{(k^*)}$ to extract features optimized for that task's classification head. We reset the SNN state (membrane potentials to zero) between these two stages to ensure temporal dynamics do not carry over from task prediction to classification each forward pass operates independently with clean initial conditions.

## A.2  DATASETS AND DATA LOADING

We conduct comprehensive evaluations on both conventional and neuromorphic datasets, namely CIFAR10, CIFAR100, Tiny-ImageNet, ImageNet100, CIFAR10-DVS, and SHD. For CIFAR10 and CIFAR100 datasets, we utilise the torchvision library to load these datasets. The corresponding link serves as an official http://cs231n.stanford.edu/tiny-imagenet-200.zip repository for the TinyImagenet dataset. We apply common data augmentations such as random cropping and horizontal flipping during training to improve generalisation.

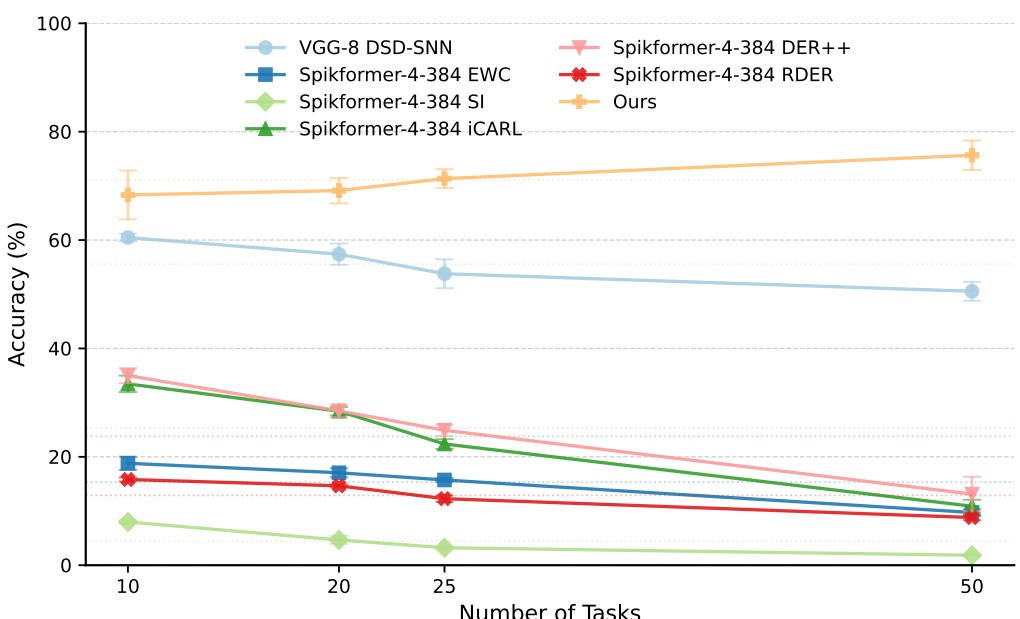

Figure 3: Reverse forgetting versus catastrophic forgetting trends on Split CIFAR-100. CATFormer (solid line) exhibits improving accuracy with increasing tasks, while baseline methods (dashed lines) show traditional forgetting patterns. Dotted lines represent average accuracy across all learned tasks.

To enable class-incremental learning, we partition each dataset into disjoint subsets forming incremental tasks. This partitioning, along with consistent label remapping, is performed within a unified `build_loaders` pipeline that returns PyTorch DataLoaders for each incremental task. We enforce reproducibility and experimental rigour by controlling all random operations including class permutations and data shuffling via fixed random seeds. This systematic approach ensures fair and consistent evaluation across different tasks and datasets.

## A.3 EXTENDED EVALUATION: PERMUTED MNIST

To demonstrate applicability beyond class-incremental settings, we evaluate CATFormer on Permuted MNIST a task-incremental benchmark with 10 sequential random permutations of the 28×28 input pixels.

Using QKFormer (pretrained) with dynamic thresholds, CATFormer achieves 90.87 ± 1.59% average accuracy across all 10 permutations. For context, classical continual learning methods report comparable performance: EWC achieves approximately 87% Kirkpatrick et al. (2017) and SI reports around 89% Zenke et al. (2017) on similar 10-task settings, though with different architectures (MLPs vs. spiking transformers).

This validates that dynamic threshold modulation extends beyond class-incremental to task-incremental scenarios, demonstrating its generality as a mechanism for continual learning.

## A.4 PLOT FOR REVERSE FORGETTING

From Figure 3, we observed a very counter-intuitive reverse forgetting trend for our method. As the number of tasks increases, the performance surpasses the mean performance across tasks, hinting at better generalization capabilities with minimal overhead. This trend we are observing for the first time, to the best of our knowledge.

## A.5 Scaling to 100 Tasks: Tiny-ImageNet Results

To test CATFormer's scalability beyond the limits of prior work, we evaluate on Tiny-ImageNet with 100 tasks, which is twice the maximum explored on CIFAR-100. Due to the absence of comparable SNN implementations on ImageNet-scale datasets, we compare against a non-spiking baseline using the Mamba architecture Gu & Dao (2023) with mixture-of-experts Yan et al. (2024); Liu et al. (2025).

As shown in Table 5, CATFormer achieves 48.56% accuracy, an 8.45% improvement over the ANN baseline (40.11%). Demonstrating dynamic threshold modulation scales effectively to handle long sequence continual learning problems.

## A.6 Neuromorphic datasets

Neuromorphic datasets present unique challenges with their spatiotemporal dynamics and event-driven nature, making them ideal testbeds for our hypothesis that dynamic thresholds align with spiking computation.

Table 5 demonstrates CATFormer's strong performance on both CIFAR10-DVS (visual events) and SHD (auditory events). On CIFAR10-DVS, we achieve 83.21%/87.14% for 2/5 task splits. Notably, our 2-task performance (83.21%) rivals the rehearsal-augmented ALADE-SNN Ni et al. (2025) (83.5%), despite using no memory buffer.

On SHD, an audio classification dataset processed with only 16 timesteps, we achieve 84.48%/87.85% for 5/10 tasks. The consistent gains validate that context-adaptive thresholds effectively exploit the temporal dimension, enabling task differentiation through firing patterns rather than explicit memory storage.

These results establish a new benchmark for rehearsal-free neuromorphic continual learning, demonstrating that our biologically-inspired mechanism naturally complements the temporal processing capabilities of SNNs.

| Dataset | Task | Best Model | | Ours |
|---|---|---|---|---|
| CIFAR10 | 5 Tasks | SA-SNN+EWC Shen et al. (2024) | $80.39 \pm 1.84$ | $\mathbf{89.29 \pm 2.53}$ |
| CIFAR10-DVS | 2 Tasks | DSD-SNN Han et al. (2023) | $80.90 \pm 1.20$ | $\mathbf{83.21 \pm 2.33}$ |
| | 5 Tasks | DSD-SNN Han et al. (2023) | $76.57 \pm 0.96$ | $\mathbf{87.14 \pm 2.78}$ |
| SHD (T=16) | 5 Tasks | DSD-SNN Han et al. (2023) | $82.56 \pm 1.15$ | $\mathbf{84.48 \pm 1.62}$ |
| | 10 Tasks | DSD-SNN Han et al. (2023) | $80.47 \pm 1.03$ | $\mathbf{87.85 \pm 1.20}$ |
| Tiny-ImageNet[3] | 100 Tasks | S6MOD Liu et al. (2025) | $40.11 \pm 0.26$ | $\mathbf{48.56 \pm 0.81}$ |

Table 5: Task-wise Average Accuracy (AA%) comparison on static (CIFAR-10, Tiny-ImageNet) and neuromorphic (CIFAR10-DVS, SHD) datasets. CATFormer achieves state-of-the-art performance across all benchmarks.

## A.7 Extended Evaluation on CIFAR-10 Dataset

A significant number of spiking continual learning methods have been evaluated on 10-class count datasets, such as MNIST or CIFAR-10 Han et al. (2023); Larionov et al. (2024); Hammouamri et al. (2022). Accordingly, we evaluate CATFormer on the CIFAR-10 dataset. The comparative results in Table 7, taken directly from Shen et al. (2024), demonstrate CATFormer's superior performance on Split CIFAR-10, achieving **83.88%** and **89.29%** average accuracy on 2 and 5-task splits, respectively. The consistent outperformance across various splits validates that our context-adaptive thresholds provide robust knowledge retention, independent of the underlying feature space dimensionality. Standard rehearsal-free SNN-based methods like SA-SNN Shen et al. (2024) and SDMLP Bricken et al. (2023), while achieving a baseline performance (**77.73%** and **73.27%** respectively on 5 tasks). Even when augmented with regularisation techniques like EWC, these methods (SA-SNN + EWC achieving **80.39%**) remain substantially below CATFormer's performance ceiling.

---

[3]S6MOD is an online continual learning method optimised for streaming data, not strictly class-incremental.

## A.8 PEFT METHODS ON QKFORMER TABLE

To examine PEFT transferability to spiking architectures, we applied various techniques to QK-Former (Table 6)

| Method | CIFAR-100 (10 tasks) |
|---|---|
| L2P | 83.18±1.20 |
| SD-LoRA | 88.01±0.31 |
| QKFormer + Identity | 18.50±1.48 |
| QKFormer + LoRA | 60.48±1.27 |
| QKFormer + SD-LoRA | 65.89±1.36 |
| QKFormer + Adapters | 72.13±0.63 |
| QKFormer + Spiking Adapters | 63.88±1.01 |
| QKFormer + Ours | **71.60±0.85** |

Table 6: PEFT technique comparison on QKFormer for CIFAR-100 (10 tasks). Dynamic threshold modulation achieves competitive performance without auxiliary architectural components.

| Model | 5 Tasks |
|---|---|
| EWC-SNN | 30.04 ±2.65 |
| MAS-SNN | 30.44 ±2.60 |
| SDMLP | 73.27 ±1.28 |
| SA-SNN(rate) | 76.88 ±2.12 |
| SA-SNN | 77.73 ±1.95 |
| FlyModel | 70.09 ±0.51 |
| SDMLP + EWC | 78.64 ±0.30 |
| SA-SNN + EWC | 80.39 ±1.84 |
| **CATFormer(Ours)** | **89.29 ± 2.53** |

Table 7: Comparative CIL accuracies on CIFAR-10 for ANN and SNN baselines vs. CATFormer.

## A.9 EVALUATION METRICS

Continual learning performance is characterised by three complementary dimensions: overall accuracy, stability of past knowledge, and plasticity for new tasks. We quantify these using standard metrics:

**Average Accuracy (AA):** Measures overall performance across all tasks after training on task $k$:

$$AA_k = \frac{1}{k} \sum_{j=1}^{k} a_{k,j} \tag{2}$$

where $a_{k,j}$ is the accuracy on task $j$ after training on task $k$. This has been reported in all the tables such as tables 7 5 1 6

**Backward Transfer (BWT):** Quantifies knowledge retention by measuring performance changes on previous tasks:

$$BWT_k = \frac{1}{k-1} \sum_{j=1}^{k-1} (a_{k,j} - a_{j,j}) \tag{3}$$

BWT implicitly captures forgetting of the model. Positive BWT indicates improved performance on old tasks (reverse forgetting), while negative BWT indicates forgetting. 0 indicates no forgetting, which is mostly hard to achieve; therefore, we consider negative but near-zero to be less forgetting and ideal in the current CIL methods.

**Forgetting Measure (FM):** Captures the maximum performance degradation on previously learned tasks:

$$FM_k = \frac{1}{k-1} \sum_{j=1}^{k-1} \max_{l \in \{j,...,k-1\}} (a_{l,j} - a_{k,j}) \tag{4}$$

| Dataset | Tasks | AA (%) | BWT (%) |
|---|---|---|---|
| *SNN Continual Learning Methods (DSD-SNN)* | | | |
| CIFAR-100 | 10 | 60.47 ± 0.72 | -10.37 |
| CIFAR-100 | 50 | 50.55 ± 1.76 | -12.88 |
| *CATFormer (SpikFormer backbone)* | | | |
| CIFAR-100 | 10 | 68.33 ± 4.51 | -4.98 |
| CIFAR-100 | 20 | 69.13 ± 2.36 | -4.87 |
| CIFAR-100 | 50 | 75.66 ± 2.72 | -5.16 |
| CIFAR-10 | 5 | 89.29 ± 2.53 | -2.39 |
| CIFAR10-DVS | 2 | 83.21 ± 2.33 | -6.28 |
| CIFAR10-DVS | 5 | 87.14 ± 2.78 | -4.85 |
| SHD (T=16) | 5 | 84.48 ± 1.62 | -5.26 |
| SHD (T=16) | 10 | 87.85 ± 1.20 | -6.55 |
| *CATFormer (QKFormer pretrained backbone)* | | | |
| CIFAR-100 | 10 | 71.92 ± 0.85 | -4.86 |
| CIFAR-100 | 50 | 77.89 ± 1.45 | -3.68 |
| Permuted MNIST | 10 | 90.87 ± 1.59 | -1.28 |

Table 8: Backward Transfer (BWT) analysis across all evaluated benchmarks. CATFormer consistently exhibits near 0 BWT, validating the effectiveness of dynamic threshold modulation for continual learning.

## A.10 ARCHITECTURAL CONTRIBUTIONS

The architectural innovation driving CATFormer's success lies in enabling task-specific modulation of neuronal firing thresholds rather than relying on static parameters. A key contribution to CATFormer's ability to mitigate catastrophic forgetting is the introduction of *dynamic, per-task firing thresholds* in spiking neurons. By enabling each task-specific head to modulate neuron firing thresholds, the model effectively partitions neuronal activations across tasks, reducing interference. This is further complemented by freezing the backbone weights after initial training on task 0, ensuring stable feature representations while task-specific thresholds and heads adapt freely.

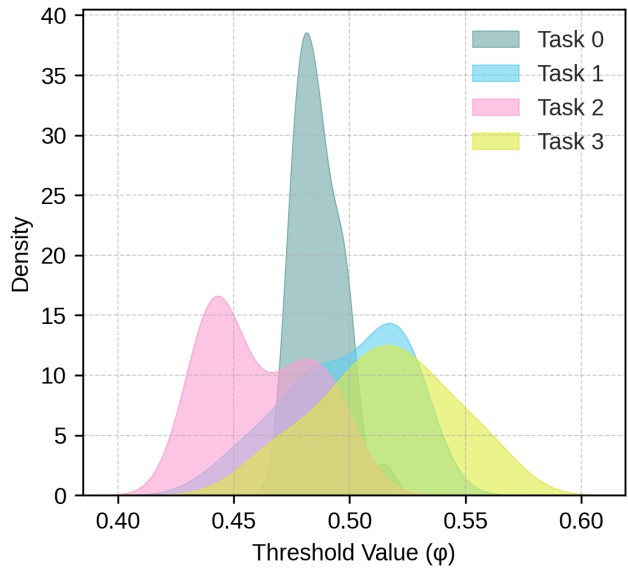

Figure 4: Depiction of threshold distribution per task.

Figure 4 describes the task-specific threshold distributions. As is obvious, the nature of the distribution will be influenced by task hardness. This dynamic partitioning creates distinct activation

patterns without interference of tasks from a biological inspiration for rehearsal-free continual learning scenarios. Each task essentially "tunes" the network's excitability profile, enabling effective task differentiation while preserving previously learned knowledge.

## A.11 HYPERPARAMETER SETTINGS PER DATASET

| Hyperparameter | CIF100 | CIF10 | C10-DVS | SHD | Tiny-IM | Im-100 |
|---|---|---|---|---|---|---|
| Batch size | 128 | 128 | 128 | 128 | 128 | 64 |
| Embedding dimension | 384 | 384 | 384 | 384 | 384 | 384 |
| Number of heads | 12 | 12 | 12 | 12 | 12 | 8 |
| MLP expansion ratio | 4 | 4 | 4 | 4 | 4 | 4 |
| Transformer blocks (depth) | 4 | 4 | 4 | 4 | 4 | 10 |
| Time steps (T) | 4 | 4 | 4 | 4 | 4 | 4 |
| Initial threshold $\phi_0$ | 0.5 | 0.5 | 0.5 | 0.5 | 0.5 | 0.5 |
| Learning rate | 3e-4 | 3e-4 | 3e-4 | 3e-4 | 1e-4 | 1e-3 |
| Epochs (initial task) | 50 | 50 | 50 | 50 | 50 | 30 |
| Epochs (incremental tasks) | 120 | 80 | 50 | 50 | 50 | 40 |
| Gating MLP learning rate | 1e-3 | 1e-3 | 1e-3 | 1e-3 | 1e-3 | 1e-3 |
| Optimizer | AdamW | AdamW | AdamW | AdamW | AdamW | AdamW |

Table 9: Hyperparameter Table

Each of the experiments is conducted thrice with different seeds on the same hyperparameters, seeds being 42, 43, and 0. The following tables describe the hyperparameters used for each dataset.

## A.12 HARDWARE AND SOFTWARE ENVIRONMENT

All experiments were conducted on NVIDIA RTX A6000 GPUs utilising CUDA 12.8 to fully leverage the available hardware acceleration. The training infrastructure comprised a workstation running Ubuntu 22.04.5 LTS with Python 3.10.12. Our implementation is based on PyTorch 2.7.1 and torchvision 0.22.1, facilitating efficient model development and execution.

To enhance training stability and computational efficiency, we employed mixed precision training via automatic mixed precision (AMP) and incorporated gradient clipping with a maximum norm of 1.0 across all optimization steps.

## A.13 ENERGY ESTIMATES

| Model | Operations | Energy (mJ) | vs. ViT |
|---|---|---|---|
| ViT-B/16 (768-dim, 12L) | $5.45 \times 10^9$ MAC | 25.07 | 1.0× |
| SpikFormer-4-384 (single)[*] | $6.46 \times 10^8$ AC | 0.58 | 43.2× |
| QKFormer-10-384 (single)[†] | $1.62 \times 10^9$ AC | 1.46 | 17.2× |
| SpikFormer-4-384 (two-pass)[*] | $1.29 \times 10^9$ AC | 1.17 | 21.4× |
| QKFormer-10-384 (two-pass) | $3.23 \times 10^9$ AC | 2.92 | 8.6× |

Table 10: Energy consumption per evaluation sample

## A.14 DTLIF IMPLEMENTATION

Our architecture leverages the SpikingJelly framework, utilising its clock-driven surrogate gradient-based LIF neuron models to simulate biologically plausible spiking dynamics over discrete time steps. We extend the classic LIF model by introducing task-conditioned, learnable firing thresholds, enabling *Dynamic Threshold LIF* neurons.

This dynamic threshold mechanism allows each channel, i.e., every neuron in a channel, to have shared thresholds that adapt its excitability per incremental task, effectively partitioning neural activations across tasks without modifying synaptic weights post-initial training. Such modulation

provides a robust strategy to mitigate catastrophic forgetting by minimizing representational overlap and promoting task-specific neural selectivity.

```python
import torch
from torch import nn
from spikingjelly.clock_driven import neuron, surrogate

class DynamicThresholdLIF(nn.Module):
    """
    MultiStepLIF with pre-defined, task-specific thresholds.
    This version pre-allocates ParameterDicts for all tasks in __init__
    to be compatible with optimizers and handles tensor reshaping.
    """
    def __init__(self, num_neurons, num_tasks, channel_dim,
                 tau=2.0, init_th=1.0, detach_reset=True):
        super().__init__()
        self.lif = neuron.MultiStepLIFNode(
            tau=tau,
            surrogate_function=surrogate.ATan(),
            detach_reset=detach_reset
        )
        self.num_tasks = num_tasks
        self.num_neurons = num_neurons
        self.channel_dim = channel_dim
        self.init_th = init_th
        self.task_th = nn.ParameterDict()
        for i in range(num_tasks):
            key = f"t{i}"
            self.task_th[key] = nn.Parameter(
                torch.full((num_neurons,), float(init_th))
            )

    def forward(self, x, task_id=None):
        if task_id is not None:
            key = f"t{task_id}"
            phi = self.task_th[key]
        else:
            phi = torch.full(
                (self.num_neurons,), self.init_th,
                device=x.device, dtype=x.dtype
            )
        shape = [1] * x.dim()
        shape[self.channel_dim] = phi.numel()
        self.lif.v_threshold = phi.view(shape).to(x.dtype)
        return self.lif(x)

    def reset(self):
        self.lif.reset()
```

Listing 1: Dynamic threshold LIF neuron implementation.

