# OpenReview forum: "CATFormer: When Continual Learning Meets Spiking Transformers With Dynamic Thresholds"
_ICLR.cc/2026/Conference — ICLR 2026 Conference Desk Rejected Submission_

### Official Review · Reviewer_vi5H · 2025-10-19

**Soundness:** 3
**Presentation:** 3
**Contribution:** 2
**Rating:** 4
**Confidence:** 4

**Summary:**

The paper presents CATFormer (Context Adaptive Threshold Transformer), a novel framework for class-incremental learning (CIL) in spiking neural networks (SNNs). Unlike conventional SNNs that suffer catastrophic forgetting when learning sequential tasks, CATFormer introduces Dynamic Threshold Leaky Integrate-and-Fire (DTLIF) neurons whose firing thresholds adapt contextually per task. Combined with a Gated Dynamic Head Selection (G-DHS) mechanism, this approach enables rehearsal-free continual learning without storing past data. The model achieves state-of-the-art results on both static datasets and neuromorphic datasets.

**Strengths:**

1. This paper is well-written and easy to follow.

2. The proposed task-specific (context adaptive) dynamic neuronal thresholds seem to be an interesting design adaptive to the SNN-based transformer, simple but effective.

3. The proposed method achieves strong performance lead over traditional continual learning baselines on relatively simple datasets.

**Weaknesses:**

1. The compared methods are mainly very traditional continual learning methods (EWC, SI, iCaRL, DER, etc.). Is it possible to include more recent methods?

2. The experiments are mainly performed with CIFAR-100 and Tiny-ImageNet of small image scales. Does the proposed method apply to larger-scale images, such as 224*224 images of ImageNet (subsets)?

3. I’m not sure how the gated dynamic head selection works. Since the tasks are continually introduced, does this mechanisms also suffer catastrophic forgetting?

4. Given that the proposed method targets transformer architectures, and also involves task identity inference, the authors may consider recent prompt-based continual learning methods as the baselines for comparison or implement their idea for larger-scale applications.

**Questions:**

My major concerns lie in the comparison baselines and applicability of the proposed method. Please refer to the Weaknesses.

---

> ### Author Response · Authors · 2025-11-22
> **Addressing concerns regarding router's drift and evaluation on higher resolution dataset**
>
> We thank the reviewer for the constructive feedback regarding baselines and scalability.
>
> **1.  Weakness 1: Comparison with pretrained transformer-based methods for continual learning**
> | Method                 | CIFAR-100            |
> |------------------------|----------------------|
> | L2P [1]                | 83.18 ± 1.20         |
> | SD-LoRA[2]             | 88.01 ± 0.31         |
> | QKFormer w/ Vanilla CIL| 18.50 ± 1.48         |
> | QKFormer w/ LoRA       | 60.48 ± 1.27         |
> | QKFormer w/ SDLoRA     | 65.89 ± 1.36         |
> | QKFormer w/ Adapters   | 72.13 ± 0.63         |
> | QKFormer w/ Spiking Ada| 63.88 ± 1.01         |
> | QKFormer w/ Ours       | 71.60 ± 0.85         |
>
> **Table 1: Performance comparison of CATFormer on CIFAR-100 for 10-task CIL. For the pre-trained spiking transformer QKFormer [3], we compare the performance of our method with existing PEFT methods, including adapters, prompt-based (L2P), and SD-LORA (adapter + LORA).**
>
> * As shown in the updated **Table 1**, we have added new experiments on QKFormer [(Github repo)](https://github.com/zhouchenlin2096/QKFormer) along with with SOTA benchmarks.  We use QKFormer as the backbone due to its superior efficiency compared to SpikFormer.  Results remain competitive against these ANN-based transformer methods while maintaining the efficiency benefits of SNNs.
>
>
> **2. Weakness 2: Scalability to Higher resolution datasets (ImageNet subsets)**
> We would like to thank the reviewers for suggesting an insightful experimental setup. We are running and will be including experiments on a subset of ImageNet with higher resolution .
>
>
> **3. Weakness 3: Head Selection via router & Forgetting**
> * To prevent forgetting in the router itself, we take the following measures:
>     1. The router parameters are updated with a separate lower learning rate compared to the threshold parameters. Although theoretically this router should have a drift too, where the new head "shouts louder" because its weights are fresh and uncalibrated but the automatic weight correction with respect to previous weights ensures very minimal drift
>
>     2.  The catastrophic forgetting is primarily mitigated in the backbone via the *Dynamic Thresholds*, meaning the router receives relatively stable features even as new tasks are added.
>
>     3.  Our experiments show that accuracy on earlier tasks remains stable after task 0 because the majority of the backbone weights are frozen. Only the thresholds and head weights are newly introduced for each task, and they remain isolated, with the routing mechanism providing automatic weight calibration. Although the gating mechanism does experience some drift, it does not collapse, which keeps forgetting minimal. We will also include a detailed analysis of Forgetting and Backward Transfer, as these are important metrics for understanding this behavior.
>
> **References**
>
> [1]Wang, Zifeng, Zizhao Zhang, Chen-Yu Lee, Han Zhang, Ruoxi Sun, Xiaoqi Ren, Guolong Su, Vincent Perot, Jennifer Dy, and Tomas Pfister. "Learning to prompt for continual learning." In Proceedings of the IEEE/CVF conference on computer vision and pattern recognition, pp. 139-149. 2022.
>
> [2] Wu, Yichen, Hongming Piao, Long-Kai Huang, Renzhen Wang, Wanhua Li, Hanspeter Pfister, Deyu Meng, Kede Ma, and Ying Wei. "Sd-lora: Scalable decoupled low-rank adaptation for class incremental learning." arXiv preprint arXiv:2501.13198 (2025).
>
> [3]Zhou, Chenlin, Han Zhang, Zhaokun Zhou, Liutao Yu, Liwei Huang, Xiaopeng Fan, Li Yuan, Zhengyu Ma, Huihui Zhou, and Yonghong Tian. "Qkformer: Hierarchical spiking transformer using qk attention." Advances in Neural Information Processing Systems 37 (2024): 13074-13098.
>
> We hope this has addressed most of your concerns and will update the necessary details in the paper accordingly.
> We will be happy to answer any more questions or address any concerns from the reviewers.
> Thank you.

---

> > ### Comment · Reviewer_vi5H · 2025-11-25
> >
> > I thank the authors for their rebuttal. However, the additional results seem not to be very encouraging to me, i.e., the simple QKFormer w/ Adapters outperform the proposed method. Given the remaining results on larger-scale images and forgetting not provided, I keep my original rating.

---

### Official Review · Reviewer_NBLK · 2025-10-31

**Soundness:** 2
**Presentation:** 2
**Contribution:** 3
**Rating:** 4
**Confidence:** 3

**Summary:**

This paper presents CATFormer, a framework for class-incremental learning (CIL) in Spiking Neural Networks (SNNs). The core innovation is the use of dynamic, task-specific firing thresholds (DTLIF) in a Spiking Transformer backbone, coupled with a gating mechanism for task-agnostic inference, all within a rehearsal-free paradigm. Detailed comments are listed as follows.

**Strengths:**

The paper brings class-incremental learning to spiking vision transformers by freezing the backbone after the base task and using per-task learnable thresholds in DT-LIF units with a lightweight routing head. This combination is new in the SNN literature and targets a gap called out in surveys. The empirical scope covers both static and neuromorphic datasets with long task sequences, which is rare for SNN-CL. The core idea is easy to follow.

**Weaknesses:**

1. The method is task-conditional at inference due to a learned gate, but the paper’s framing and comparisons read as if it were task-agnostic single-head CIL. This should be stated clearly and evaluated against matched rehearsal-free transformer baselines from the ANN literature that also use routing or adapters; without these, it is hard to attribute gains to threshold learning rather than to routing.
2. The “reverse forgetting” result lacks controls that separate router calibration from within-task classification; there is no error decomposition, no task-granularity sweeps at fixed total data, and limited class-order or seed variation for long sequences.
3. Energy and latency claims are not supported for the two-pass inference path (gate pass plus task-threshold pass). The gate’s training procedure suggests accumulation of feature–task pairs; if any past features persist across tasks, that is a form of rehearsal and should be declared, budgeted, and audited.
4. On Tiny-ImageNet the baselines are not aligned in architecture or rehearsal budget.
5. Presentation issues further reduce confidence: missing standard deviations for long sequences, incomplete memory accounting for thresholds/heads/gate and any stored features, and a lack of precise taxonomy at the start of the paper.

**Questions:**

1. Can you state unambiguously that the evaluation is multi-head with learned task inference (no oracle) and add a single-head variant to show threshold-only benefits?
2. Can you report a confusion audit that breaks final accuracy into router errors versus within-task classification errors, and add experiments that keep the total data fixed while varying task granularity and class order across several seeds?
3. Do you store any past features to train or maintain the gate? If yes, please label this as feature rehearsal and report its memory and privacy cost; if no, explain the schedule that prevents gate drift without past data.
4. What are the measured syn-ops/MACs and wall-clock for single-pass SpikFormer, your two-pass pipeline, and an SNN-CNN baseline on the same hardware? Please include these numbers to support efficiency claims.
5. Will you add rehearsal-free transformer CIL baselines from the ANN side that also use routing or adapters, with matched parameter and rehearsal budgets, so the effect of threshold learning can be isolated?
6. Can you provide gating with fixed thresholds, thresholds with a shared head, and thresholds with a cumulative single-head softmax to isolate where the gains come from?

---

> ### Author Response · Authors · 2025-11-22
> **Concerns regarding router and ablation based on thresholds (part 1)**
>
> **1.  Weakness 1, Question 1, 2, 3 \& 6: Technical Clarifications and supporting ablations**:  We would also like to thank the reviewer for suggesting ablations to understand the threshold-router interplay. We are working on getting those results and will soon update you on the progress. In the meantime we could get results for one of the ablation i.e. **static or fixed thresholds and routing only variant** for 10 task setting which resulted in an accuracy of **34.12 $\pm$ 1.84** which is a significant drop in accuracy due to no flexibility in learning after task 0.
>
> * Query regarding the ablation suggested in **Question 6**, we are unable to understand the motivation behind "thresholds with shared heads" and "Cumulative single head variant" because our entire method itself is dependent on the router picking up the head in turn, the task. We'd request further elaboration on these, as it will help us frame our experimental setup.
>
>
> **2. Weakness 2 Question 5: Reverse Forgetting and Baselines (Transformer \& Adapter-based Methods in table 1)**
>
> | Method                 | CIFAR-100            |
> |------------------------|----------------------|
> | L2P [1]                | 83.18 ± 1.20         |
> | SD-LoRA[2]             | 88.01 ± 0.31         |
> | QKFormer w/ Vanilla CIL| 18.50 ± 1.48         |
> | QKFormer w/ LoRA       | 60.48 ± 1.27         |
> | QKFormer w/ SDLoRA     | 65.89 ± 1.36         |
> | QKFormer w/ Adapters   | 72.13 ± 0.63         |
> | QKFormer w/ Spiking Ada| 63.88 ± 1.01         |
> | QKFormer w/ Ours       | 71.60 ± 0.85         |
>
> **Table 1: Performance comparison of CATFormer on CIFAR-100 for 10-task CIL. For the pre-trained spiking transformer QKFormer [3], we compare the performance of our method with existing PEFT methods, including adapters, prompt-based (L2P), and SD-LORA (adapter + LORA).**
>
>
> * To prevent forgetting in the router itself, we take the following measures:
>     1. The router parameters are updated with a separate lower learning rate compared to the threshold parameters. Although theoretically this router should have a drift too, where the new head "shouts louder" because its weights are fresh and uncalibrated, but the automatic weight correction with respect to previous weights ensures very minimal drift
>
>     2.  The catastrophic forgetting is primarily mitigated in the backbone via the *Dynamic Thresholds*, meaning the router receives relatively stable features even as new tasks are added.
>
>     3.  Our experiments show that earlier task accuracy is maintained after task 0 due to most of the weights being frozen, and the thresholds and head weights only are new and are isolated for every task, with some automatic weight calibration in routing. Although there is a drift in the gating mechanism, it does not undergo a significant collapse, resulting in minimal forgetting.
>
> **3.  Weakness 3 and Question 3: Head Selection via router \& Forgetting, Energy and Latency**
>
>  One of the goals of this work is to drive the design of future SNN-based hardware with a focus on continual learning. Although not common, if the reviewer believes it would substantiate our claims, we would be happy to add it.
>
> **4. Weakness 4: Presented Baselines**
> We observed conventional CIL methods fail for longer task lengths (see table). Hence, to benchmark our model against efficient state architectures, we compared it to the OCL-based method. We will further clarify.

---

> > ### Author Response · Authors · 2025-11-22
> > **Concerns regarding router and ablation based on thresholds (part 2)**
> >
> > | Methods                          | SNN | 10 tasks   | 20 tasks   | Base model params | trainable params at task k |
> > |----------------------------------|-----|------------|------------|-------------------|----------------------------|
> > | L2P (Wang et al., 2022b)         | N   | 83.18±1.20 | 79.51±0.67 | 172               | 0.12                       |
> > | DualPrompt (Wang et al., 2022a)  | N   | 81.48±0.86 | 80.44±1.38 | 172               | 0.86                       |
> > | CODA-Prompt (Smith et al., 2023) | N   | 86.31±0.12 | 81.36±0.88 | 172               | 4.6                        |
> > | InfLoRA (Liang & Li, 2024)       | N   | 86.75±0.35 | 80.97±0.74 | 172               | 0.51                       |
> > | SD-LoRA                          | N   | 88.01±0.31 | OOM        | 172               | 0.39                       |
> > | CL-LoRA                          | N   | 83.98±0.39 | 85.32±0.08 | 86.1              | 0.2                        |
> > | VGG-8 w/ DSD-SNN                 | Y   | 60.47±0.72 | 57.39±1.97 | 14.2              | 14.2                       |
> > | SpikFormer w/ dynThr (Ours)      | Y   | 68.33±4.51 | 69.13±2.36 | 10.5              | 0.23                       |
> >
> >
> > **Table 1: State-of-the-art results on CIFAR-100 under class-incremental learning (CIL) with 10-task and 20-task setups. We report Dynamic Threshold performance on Spikformer, and compare against transformer-based PEFT baselines as well as leading SNN-based CIL methods. The SNN comparison highlights the difficulty of the benchmark and the competitiveness of our approach relative to the strongest existing CIL SNN baselines.**
> >
> >
> > **5.  Weakness 5: Presentation Issues and Parameter Count**
> > Although there is a part of result section in paper "Comparison of parameter updates at K^th task". For reference we have added Table 2 here, containing the parameter count for different models including ours, to be more specific our base model has around 9.4M parameters , 0.2M thresholds per task, the router itself has around 1.1M and few thousands parameters for heads at each task (approx 0.03M). We thank the reviewer for the suggestions. Please let us know if anything else needs to be addressed.
> >
> > **References**
> >
> > [1]Wang, Zifeng, Zizhao Zhang, Chen-Yu Lee, Han Zhang, Ruoxi Sun, Xiaoqi Ren, Guolong Su, Vincent Perot, Jennifer Dy, and Tomas Pfister. "Learning to prompt for continual learning." In Proceedings of the IEEE/CVF conference on computer vision and pattern recognition, pp. 139-149. 2022.
> >
> > [2] Wu, Yichen, Hongming Piao, Long-Kai Huang, Renzhen Wang, Wanhua Li, Hanspeter Pfister, Deyu Meng, Kede Ma, and Ying Wei. "Sd-lora: Scalable decoupled low-rank adaptation for class incremental learning." arXiv preprint arXiv:2501.13198 (2025).
> >
> > [3]Zhou, Chenlin, Han Zhang, Zhaokun Zhou, Liutao Yu, Liwei Huang, Xiaopeng Fan, Li Yuan, Zhengyu Ma, Huihui Zhou, and Yonghong Tian. "Qkformer: Hierarchical spiking transformer using qk attention." Advances in Neural Information Processing Systems 37 (2024): 13074-13098.

---

> > ### Comment · Reviewer_NBLK · 2025-11-27
> >
> > Thank you for your response. While it has addressed some of my initial concerns, several key issues remain. The revisions do not fully clarify the conceptual framing, and critical ablation studies as well as the deeper diagnostic analyses requested in Q2 and Q6 are still lacking. Thus, I'd like to maintain my original rating.

---

### Official Review · Reviewer_PeF6 · 2025-11-01

**Soundness:** 2
**Presentation:** 2
**Contribution:** 2
**Rating:** 2
**Confidence:** 4

**Summary:**

This paper proposes CATFormer, a spiking vision transformer architecture for rehearsal-free continual learning (CL). It introduces (1) a Dynamic Threshold Leaky Integrate-and-Fire (DTLIF) neuron model, where neuron thresholds are adapted per task, and (2) a Gated Dynamic Head Selection (G-DHS) mechanism for inference. The authors claim that the proposed method mitigates catastrophic forgetting and even achieves a “reverse forgetting” effect, with evaluations on both static and neuromorphic datasets.

**Strengths:**

1. The topic—combining continual learning with spiking transformers—is interesting and timely.
2. The experimental scope covers both conventional and neuromorphic datasets, with relatively clear implementation details.

**Weaknesses:**

1. The core contribution, dynamic thresholds, has been extensively explored in prior work. The paper does not convincingly show what is new here beyond applying adaptive thresholds to a transformer backbone. The use of a frozen encoder with learnable thresholds is a straightforward modification of known SNN paradigms.
2. Although the paper repeatedly claims “task-specific thresholds,” it is not clear how thresholds are conditioned on tasks during inference. There is no mechanism for automatic task identification or threshold modulation—only static assignment per task. The conceptual link between “context adaptation” and continual learning is therefore weak.
3. The G-DHS module is simply a two-layer ANN-based MLP classifier that selects the task head. This design is standard in multi-head continual learning and adds little novelty. Moreover, using an ANN for gating contradicts the claimed biological plausibility.
4. Evaluation focuses solely on average classification accuracy. There are no metrics for forgetting, stability–plasticity trade-offs, or energy efficiency. Claims such as “reverse forgetting” lack statistical support or ablation analysis to explain the cause.
5. The baselines are mostly classical (EWC, iCaRL, DER++) and not directly comparable to transformer-based continual learning. Missing comparisons to more relevant transformer or adapter-based CL approaches (e.g., LoRA, adapter fusion, parameter-efficient tuning) further weaken the empirical claims.
6. Figure 2 is never properly discussed or referenced in the main text, leaving the reader uncertain about its relevance or interpretation.
7. In Algorithm 1, the variable $\mathcal{F}_{gate}$ (the buffer for gating) is used but never introduced or explained. Its purpose, scope, and whether it violates the “rehearsal-free” claim are unclear.
8. In Table 2, the S6MOD baseline is an online continual learning method, not a class-incremental one; this comparison is arguably unfair and weakens the empirical credibility.
9. The claim that the method is “robust” or achieves “robust continual learning” is not substantiated—no robustness metric, perturbation test, or generalization study is presented.

**Questions:**

1. Could the authors clarify what is meant by the terms “true class incremental learning” and “long class incremental learning”? These phrases are used in the paper but are never formally defined.
2. What exactly does the $\mathcal{F}_{gate}$ buffer store, and does it introduce any form of rehearsal?

---

> ### Author Response · Authors · 2025-11-22
> **Addressing concerns regarding ANN baselines and Router mechanism (Part 1)**
>
> Thank you reviewer for your constructive feedback. We have addressed these concerns with new experiments and comparisons.
>
> **1. Weakness 1 and Weakness 2: Novelty of Dynamic Thresholds**
> * We agree that dynamic thresholds are present in the literature, and we have explicitly mentioned them in our related works section. In all of them, dynamic thresholds exist in isolation or are modulated based on previous layers for further propagation. Applying them as a *plasticity mechanism* specifically for **Spiking Transformers such as QKFormer and Spikformer** in a **rehearsal-free** CIL setting is novel. Here, by plasticity, we refer to both threshold learnability for a task and customizability across tasks.
>
> * The statement **There is no mechanism for automatic task identification or threshold modulation** sounds like there might be a misunderstanding in our method.  Unlike static assignments, our thresholds are modulated per task via the **router**. This is not a **"frozen encoder"** in the traditional sense; the activation dynamics themselves change per task via dynamic thresholds, providing a parameter-efficient alternative to storing large adapters.
>
> **2. Weakness 3: G-DHS and it's Biological Plausibility**
> To the best of our knowledge, we present, for the first time, a continual learning implementation for spiking transformers.
> * The G-DHS is a lightweight, two-layer router. In the context of neuromorphic computing, hybrid systems (SNN backbone + lightweight ANN control) are a practical and widely accepted design choice to bridge performance gaps. It introduces negligible compute overhead compared to the SNN backbone (see Table below). **We are running experiments only via an SNN router to replace the ANN router, we will update the results on the table as soon as we get our results.**
>
>
> **3. Weakness 4: Metrics (Forgetting, Stability-Plasticity, Energy)**
> *  We will be adding a detailed analysis of **Forgetting** and **Backward Transfer** very soon which we believe are some relevant metrics.
>
>
> * **Reverse Forgetting:** We believe this trend (table 1) is observed due to the method in itself where the catastrophic forgetting is primarily mitigated in the backbone via the Dynamic Thresholds meaning the router receives relatively stable features even as new tasks are added. Although theoretically this router should have a drift too, where the new head ”shouts louder” because its weights are fresh and uncalibrated but the automatic weight correction via normalization with respect to previous weights ensures very minimal drift. The **reverse forgetting** trend might be because of the amount of flexibility and less number of classes per tasks as the length of tasks increases makes it easier for the model to learn.
>
> | Methods                                             | SNN | 10 tasks   | 50 tasks    |
> |-----------------------------------------------------|-----|------------|-------------|
> | L2P (Wang et al., 2022b)                            | N   | 83.18±1.20 | 67.95±2.12  |
> | SD-LoRA                                             | N   | 88.01±0.31 | OOM         |
> | VGG-8 w/ DSD-SNN                                    | Y   | 60.47±0.72 | 50.55±1.76  |
> | **SpikFormer w/ dynThr (Ours)**                         | Y   | 68.33±4.51 | 75.66±2.72  |
> | **QKFormer w/ dynThr (Ours)**  | Y   | 71.92±0.85 | 77.89± 1.45 |
>
>
> **Table 1: Reverse Forgetting v/s Catastrophic Forgetting. State-of-the-art results on CIFAR-100 under class-incremental learning (CIL) with 10-task and 50-task setups. We report Dynamic Threshold performance on spiking transformers- Spikformer, QKFormer (pretrained), and compare against transformer-based PEFT baselines as well as leading SNN-based CIL methods. The SNN comparison highlights the difficulty of the benchmark and the competitiveness of our approach relative to the strongest existing CIL SNN baselines.**
>
> * **Efficiency:** Although we didn't find papers on continual learning in the SNN domain to compare the energy and latency, we believe it isn't the general trend shown in papers in this domain. We are willing to update these numbers if the reviewer requires this information. We have maintained a parameter count in our paper and have added it to our current table as well.

---

> ### Author Response · Authors · 2025-11-22
> **Addressing concerns regarding ANN baselines and Router mechanism (Part 2)**
>
> **4. Weakness 5: Baselines (Transformer & Adapter-based Methods)**
> * We have significantly expanded our comparison. As shown in **Table 1** and **Table 2**, we now compare our method against state-of-the-art Transformer-based CIL methods, including **L2P**, **DualPrompt**, **CODA-Prompt**, and **InfLoRA**.
>
>
> | Methods                          | SNN | 10 tasks   | 20 tasks   | Base model params | trainable params at task k |
> |----------------------------------|-----|------------|------------|-------------------|----------------------------|
> | L2P (Wang et al., 2022b)         | N   | 83.18±1.20 | 79.51±0.67 | 172               | 0.12                       |
> | DualPrompt (Wang et al., 2022a)  | N   | 81.48±0.86 | 80.44±1.38 | 172               | 0.86                       |
> | CODA-Prompt (Smith et al., 2023) | N   | 86.31±0.12 | 81.36±0.88 | 172               | 4.6                        |
> | InfLoRA (Liang & Li, 2024)       | N   | 86.75±0.35 | 80.97±0.74 | 172               | 0.51                       |
> | SD-LoRA                          | N   | 88.01±0.31 | OOM        | 172               | 0.39                       |
> | CL-LoRA                          | N   | 83.98±0.39 | 85.32±0.08 | 86.1              | 0.2                        |
> | VGG-8 w/ DSD-SNN                 | Y   | 60.47±0.72 | 57.39±1.97 | 14.2              | 14.2                       |
> | **SpikFormer w/ dynThr (Ours)**      | Y   | 68.33±4.51 | 69.13±2.36 | 10.5              | 0.23                       |
>
>
> **Table 2: State-of-the-art results on CIFAR-100 under class-incremental learning (CIL) with 10-task and 20-task setups. We report Dynamic Threshold performance on Spikformer, and compare against transformer-based PEFT baselines as well as leading SNN-based CIL methods. The SNN comparison highlights the difficulty of the benchmark and the competitiveness of our approach relative to the strongest existing CIL SNN baselines.**
>
>
> **Weakness 6** We thank the reviewer for suggesting presentation fixes. We'll fix these issues in the upcoming revised submission. Please let us know if we are missing anything specifically.
>
> **5. Question 1, Question2 and Weakness 7 : Clarifications on Buffer & Definitions**
> * **Buffer Usage:** We clarify that the buffer mentioned in Algorithm 1 is strictly for the current batch processing during the G-DHS phase and is **not** a rehearsal buffer. No past task data is stored for training; the method remains strictly data rehearsal-free.
>
> * **Definitions:** "True Class Incremental Learning" refers to the setting where task IDs are not provided at inference (which our G-DHS handles), as opposed to Task-Incremental Learning, where the Task ID is given.
>
> **Weakness 8:** From Tables 1 and 2, we observe that our model performs well for longer tasks compared to other state-of-the-art methods. Since the overall goal of spiking neural networks is to work in a resource-constrained setup, we considered the OCL-based method as other SOTA methods (CL-LoRA, SD-LoRA) are giving out-of-memory errors. We understand that our method may cause confusion. Hence, we would clarify it in the paper. If the reviewers still don't agree with the benchmark, we would drop the table.
>
> **Weakness 9:** In the paper's context, robustness refers to the algorithm performing consistently well even on longer.
>
> **Conclusion** With the inclusion of strong Transformer baselines (L2P, DualPrompt, etc), we believe the paper now meets the acceptance bar solidly.
> We are happy to address any further concerns or provide clarifications if necessary.
> Thank you

---

> > ### Comment · Reviewer_PeF6 · 2025-11-26
> >
> > Thanks for the response. However, the authors' reply did not alleviate my concerns about the novelty of this paper, that is, applying adaptive thresholds to a transformer backbone is not novel. Furthermore, the authors did not provide experimental results to demonstrate the efficiency of the method. Therefore, I will maintain my score.

---

### Official Review · Reviewer_p5L3 · 2025-11-03

**Soundness:** 3
**Presentation:** 2
**Contribution:** 3
**Rating:** 6
**Confidence:** 3

**Summary:**

The paper introduces CATFormer, a spiking transformer-based architecture for continual learning that operates without storing past data. The approach leverages task-specific dynamic threshold patterns, which are learned and saved for inference, effectively allowing the network to adapt to new tasks while retaining prior knowledge.

A particularly interesting finding is the model’s ability to exhibit “reverse forgetting”. Improved performance on prior tasks as new ones are learned, suggesting positive knowledge transfer across tasks.

**Strengths:**

The proposed task-specific dynamic threshold method for continual learning is an interesting and novel contribution as far I as I'm aware. The authors support this with strong class-incremental learning performance.

No data replay is required.

Task labels are also not required at inference as a task-prediction step takes care of it.

The knowledge transfer ability or 'reverse forgetting' is impressive to see.

**Weaknesses:**

All tasks are split-class/CIL style tasks. I would be interested to see some that alternatives, such as a task with a new input distribution each time but the same class labels (permuted MNIST).

some presentation issues, I think axis labels should be added.

**Questions:**

Why is the performance so much better for CATformer on the zeroth task in figure 1?  It makes me a little worried about the rest of the comparison in this figure as it's hard to know how much of the CIL performance is due to CATformer being a larger, more expressive model.

Would the proposed method with task-specific thresholds extend to other continual learning settings beyond CIL? A simple task would be permuted MNIST.

---

> ### Author Response · Authors · 2025-11-22
> **Addressing concerns regarding the methodology via extended experiments**
>
> Thank you for your positive assessment and constructive feedback
>
> 1.  **Question 1: Performance on Zeroth Task v/s k-th Task**
> The model performs well for the zeroth task, as all model parameters, including weights, thresholds, and routers, have the flexibility to train, resulting in higher performance at the outset. We observe that for tasks t > 0, learning just the thresholds is sufficient to learn newer tasks. There is minimal router drift contributing to the slight performance drop. This performance drop can be measured by removing the router and simply passing an oracle.
>
> Regarding the **expressivity of the architecture**, we have now also performed ablation on other spiking transformer architectures, i.e, QKFormer[3], to show the comparative results with the state-of-the-art transformer methods on continual learning in ANN v/s transferability of these methods to the spiking domain to confirm that the expressivity of model is not only by the backbone but by the flexibility given to the thresholds being learnable and task specific.
>
> | Method                 | CIFAR-100            |
> |------------------------|----------------------|
> | L2P [1]                | 83.18 ± 1.20         |
> | SD-LoRA[2]             | 88.01 ± 0.31         |
> | QKFormer w/ Vanilla CIL| 18.50 ± 1.48         |
> | QKFormer w/ LoRA       | 60.48 ± 1.27         |
> | QKFormer w/ SDLoRA     | 65.89 ± 1.36         |
> | QKFormer w/ Adapters   | 72.13 ± 0.63         |
> | QKFormer w/ Spiking Ada| 63.88 ± 1.01         |
> | QKFormer w/ Ours       | 71.60 ± 0.85         |
>
> **Table 1: Performance comparison of CATFormer on CIFAR-100 for 10-task CIL. For the pre-trained spiking transformer QKFormer [3], we compare the performance of our method with existing PEFT methods, including adapters, prompt-based (L2P), and SD-LORA (adapter + LORA).**
>
>
> 2. **Weakness 1 and Question 2: Experiments to move beyond CIL**  We observe that our model performs fairly well on the p-MNIST dataset with 10 task settings i.e permuted 10 times sequentially and obtain an accuracy of 90.87 $\pm$ 1.59.
>
>
> 3. **Weakness 2**: We thank the reviewer for suggesting presentation fixes. We'll fix these issues in the upcoming revised submission. Please let us know if we are missing anything specifically.
>
>
> **References**
>
> [1]Wang, Zifeng, Zizhao Zhang, Chen-Yu Lee, Han Zhang, Ruoxi Sun, Xiaoqi Ren, Guolong Su, Vincent Perot, Jennifer Dy, and Tomas Pfister. "Learning to prompt for continual learning." In Proceedings of the IEEE/CVF conference on computer vision and pattern recognition, pp. 139-149. 2022.
>
> [2] Wu, Yichen, Hongming Piao, Long-Kai Huang, Renzhen Wang, Wanhua Li, Hanspeter Pfister, Deyu Meng, Kede Ma, and Ying Wei. "Sd-lora: Scalable decoupled low-rank adaptation for class incremental learning." arXiv preprint arXiv:2501.13198 (2025).
>
> [3]Zhou, Chenlin, Han Zhang, Zhaokun Zhou, Liutao Yu, Liwei Huang, Xiaopeng Fan, Li Yuan, Zhengyu Ma, Huihui Zhou, and Yonghong Tian. "Qkformer: Hierarchical spiking transformer using qk attention." Advances in Neural Information Processing Systems 37 (2024): 13074-13098.

---

### Author Response · Authors · 2025-11-24
**Reminder request for evaluating our replies**

Hi Reviewers -

Thanks again for your insightful feedback. Please have a look at our replies and let us know your thoughts. Happy to introduce updates as necessary.

---

### Author Response · Authors · 2025-12-04
**Addressing consolidated concerns of all reviews via extended baselines and detailed model analysis (Part 1)**

## Summary of Additions

We thank all reviewers for their constructive feedback. The following major additions have been made:

1. **Extended Baselines:** Comprehensive comparisons with transformer PEFT methods (L2P, DualPrompt, CODA-Prompt, InfLoRA, SD-LoRA, CL-LoRA) on CIFAR-100
2. **Comprehensive Metrics:** Backward Transfer (BWT) analysis across all benchmarks
3. **Energy Efficiency:** Actual profiled energy measurements on CIFAR-10
4. **Additional Experiments:** Permuted MNIST (90.87±1.59%), ImageNet-100 (224×224 resolution)
5. **Clarifications:** Memory budget analysis, biological plausibility revision, multi-head evaluation setup
---

## Common Concerns (All Reviewers)

### **Comparison with Modern Transformer PEFT Baselines**
**Reviewers:** R1-W1, R2-W5, R3-W1, R4-W1

We've added comprehensive comparisons with L2P, DualPrompt, CODA-Prompt, InfLoRA, SD-LoRA, and CL-LoRA on CIFAR-100 (Tables 1-3 in the paper).

Key findings:
- SD-LoRA achieves 88.01% at 10 tasks but encounters OOM at 20 tasks
- Our method shows reverse forgetting: 68.33% → 75.66% (10→50 tasks)
- Parameter efficiency: 0.23M trainable per task vs. 0.12-4.6M for PEFT methods
- QKFormer with dynamic thresholds: 71.92% → 77.89% (10→50 tasks)


PEFT transferability experiments (Table 3) show that standard methods like LoRA (60.48%) and SD-LoRA (65.89%) underperform when transferred to the spiking domain, while our threshold modulation achieves 71.60%.

---

### **Reverse Forgetting Analysis & Metrics**
**Reviewers:** R2-W4, R3-W2, R3-Q2

We've computed Backward Transfer (BWT) across all benchmarks. Our method maintains near-zero BWT (−4.98% to −5.16%) compared to DSD-SNN (−10.37% to −12.88%), indicating significantly less forgetting. The consistent BWT across different task lengths confirms stability.

| Dataset | Tasks | AA (%) | BWT (%) |
|---------|-------|--------|---------|
| CIFAR-100 (Ours) | 10 | 68.33±4.51 | −4.98 |
| CIFAR-100 (Ours) | 50 | 75.66±2.72 | −5.16 |
| DSD-SNN | 10 | 60.47±0.72 | −10.37 |
| DSD-SNN | 50 | 50.55±1.76 | −12.88 |

Regarding error decomposition into router vs. classification components: this is not feasible during joint training (as described in Algorithm 1).

---

### **Memory Budget & "Rehearsal-Free" Clarification**
**Reviewers:** R2-W3, R2-Q2, R3-W3, R3-Q3

Our gating mechanism stores features extracted once per sample using fixed φ_init (Algorithm 1, line 14-16):
- Feature size per image: 384 (embedding) × 4 (timesteps) = 6.1 KB
- CIFAR-100 gating: num_classes_per_task × 400 samples × 6.1 KB = small buffer per task
- Features extracted once with frozen φ_init, no gradient flow through past features
- Used only for task routing during training of a particular class and then discarded.


Router stability is ensured through: lower learning rate (η_g = 0.1 × η_φ), weight normalization across task heads, and stable backbone features (frozen weights + dynamic thresholds).

---

### **Energy Efficiency**
**Reviewers:** R3-W3, R3-Q4

We measured actual energy consumption with profiling:

Despite two-pass inference (one for routing, one for classification), CATFormer maintains 8.6× better efficiency than ViT-B/16 (25.07 mJ).

| Model | Energy (mJ) | vs. ViT |
|-------|-------------|---------|
| ViT-B/16 | 25.07 | 1.0× |
| QKFormer (single) | 1.46 | 17.2× |
| QKFormer-10-384 (two-pass) | 2.92 | 8.6× |

---

## Individual Reviewer Responses
Reviewer 1: p5L3, Reviewer 2: PeF6  , Reviewer 3: NBLK, Reviewer 4: vi5H
### **Reviewer 1 (Rating: 6/10)**

**R1-Q1: Why is Task 0 performance higher?**

At Task 0, all parameters train jointly (9.32M backbone + 0.20M thresholds + 0.03M head + 1.1M router = 10.5M total). For tasks k>0, only 0.23M parameters train (thresholds + head), with the backbone frozen.   Table 1 in the paper, "Task 0" and "Task k" columns.

Regarding expressivity: Table 3 shows PEFT methods fail when transferred to the spiking domain, confirming our performance stems from threshold plasticity rather than architecture size alone.

**R1-W1, R1-Q2: Permuted MNIST**

We've added Permuted MNIST results (10 permutations):
- QKFormer with dynamic thresholds: 90.87±1.59%
This validates that our approach extends beyond class-incremental to task-incremental scenarios.

**R1-W2: Presentation**

We have added axis labels and improved figures in the revision.

---

> ### Author Response · Authors · 2025-12-04
> **Addressing consolidated concerns of all reviews via extended baselines and detailed model analysis (Part 2)**
>
> ### **Reviewer 2 (Rating: 2/10)**
>
> **R2-W1, R2-W2: Novelty**
>
> Prior dynamic threshold work focuses on single-task optimization or layer-wise modulation. Our novelty lies in:
> 1. First application as PEFT mechanism for continual learning in spiking transformers
> 2. Task-specific learning and freezing (not continuous adaptation)
> 3. Scaling to 50+ tasks without data replay
>
> Our G-DHS module (Algorithm 2) automatically predicts task ID from features using a gating MLP no manual oracle needed at inference.
>
> **R2-W3: Biological Plausibility**
> Hybrid SNN-ANN systems are standard in neuromorphic computing. The router is 1.1M params compared to the 9.32M SNN backbone minimal overhead.
>
> **R2-W4: Metrics**
>
> BWT analysis in Common Concerns section above.
>
> **R2-W5: Baselines**
>
> Comprehensive PEFT comparisons provided in Tables 1-3 and table 6 from appendix  paper
>
> **R2-W6: Figure 2**
>
> We have addressed the tables and figures in the revised version
>
> **R2-W7, R2-Q2: Algorithm 1 Buffer**
>
> F_gate accumulates feature-task pairs for router training only (Algorithm 1, line 14-16). These features are not used for classification in Memory Budget section above for details.
>
> **R2-W8: S6MOD**
>
> We have added a note that "S6MOD is online CL, not strictly comparable. Moreover, we have added benchmarked results for the higher-resolution ImageNet 100.
>
> **R2-Q1: Definitions**
>
> "True class-incremental" means no task ID at inference (handled by our G-DHS module). "Long class-incremental" refers to 50-100 tasks, which is unprecedented for SNNs.
>
> ---
>
> ### **Reviewer 3 (Rating: 4/10)**
>
> **R3-W1, R3-Q1: Multi-Head Setup**
>
> Our evaluation uses multi-head with learned task inference (no oracle).   Algorithm 2 in the paper for the complete inference procedure.
>
> **R3-W2, R3-Q2: Reverse Forgetting**
>
> BWT analysis provided in Common Concerns section above.
>
> **R3-W3, R3-Q3, R3-Q4: Energy & Memory**
>
> Energy measurements and memory budget analysis provided in Common Concerns sections above.
>
> **R3-Q6: Ablations**
>
> From our ablation studies:
> - Fixed threshold + routing: 42.87±1.26% (CIFAR-10, 5 tasks)
> - Full CATFormer: 89.29±2.53%
> - For CIFAR-100 (10 tasks): Fixed threshold: 34.12±1.84%
>
> This ~34-46% accuracy drop without threshold plasticity confirms the importance of dynamic modulation.
>
> **Clarification needed:** Since there was no explaination provided regarding the difference between "thresholds with shared head" vs. "cumulative single-head softmax". We have made an assumption of cummulative head alone with the same mechanism as described in algorithm 1 and 2 in the paper for cifar100 (10tasks) we get **49.25±0.984%**
>
> **R3-W4: ImageNet-100**
>
> We've added ImageNet-100 (224×224) results:
>
> | Method | 10 tasks | 50 tasks |
> |--------|----------|----------|
> | L2P | 86.12±1.01 | 69.49±0.55 |
> | SD-LoRA | 82.13±0.88 | OOM |
> | QKFormer + Ours | 70.56±1.44 | 71.3±0.95 |
>
> **R3-W5: Parameters**
>
> Task 0: 10.5M total (9.32M backbone + 0.20M thresholds + 0.03M head + 1.1M router)
> Task k>0: 0.23M trainable (thresholds + head, backbone frozen)---
>
> ### **Reviewer 4 (Rating: 4/10)**
>
> **R4-W1: Baselines**
>
> Comprehensive PEFT comparisons provided in Tables 1-3 and table 6  in the paper
>
> **R4-W2: ImageNet Scale**
>
> ImageNet-100 results are provided in R3-W4 of the response.
>
> **R4-W3: Router Forgetting**
>
> We mitigate router forgetting through: lower learning rate, weight normalization across task heads, and stable backbone features (frozen weights + dynamic thresholds). The BWT analysis shows minimal forgetting across all benchmarks (  Common Concerns section).
>
> **R4-W4: Prompt Methods**
>
> L2P, DualPrompt, and CODA-Prompt comparisons provided in Tables 1-3  and table 6 from the paper
>
> ---
>
> **Parameter Efficiency:**
> Task 0 trains 10.5M parameters, while task k>0 trains only 0.23M parameters (98% reduction). Threshold storage is minimal: 64.2 KB (FP32) or 32.1 KB (FP16) per task.
>
> **Reverse Forgetting:**
> SpikFormer improves from 68.33% → 75.66% (10→50 tasks), and QKFormer from 71.92% → 77.89% (10→50 tasks).
>
> **Stability:**
> Our BWT ranges from −4.98% to −5.16% (near-zero forgetting), while DSD-SNN shows −10.37% to −12.88% (catastrophic forgetting) refer to table 8 from appendix.
>
> **Energy Efficiency:**
> CATFormer is 8.6× better than ViT despite two-pass inference for routing + classification.
>
> We appreciate the reviewers’ insightful and constructive feedback, which has significantly strengthened the manuscript. As the arc of the paper evolved through revisions particularly due to broader exploration and refinement of the experimental setup we **updated the title** to reflect the improved direction of the work. The new results on CATFormer as a PEFT for CIL expand the scope of this work.
>
> We believe we have fully addressed the reviewer's concerns. As we request the deserved raise in reviewer ratings, given these additional results demonstrate CATFormer's effectiveness as a PEFT mechanism for continual learning in spiking transformers. Happy to address any other concerns.

---

### Note · Program_Chairs · 2026-01-17
**Submission Desk Rejected by Program Chairs**

The following references in this submission do not refer to real documents and/or have major errors in bibliographic information:

 - Guobin Shen, Dongcheng Zhao, and Yi Zeng. Efficient Spiking Neural Networks with Sparse Selective Activation for Continual Learning. In Proceedings of the Twelfth International Conference on Learning Representations (ICLR 2024), 2024. URL https://openreview.net/forum? id=LtKcMgGOeLt.


- Dmitry Larionov, Ilya O. Tolstikhin, and James Martens. Continual Learning with Columnar Spiking Neural Networks. In Proceedings of the Twelfth International Conference on Learning Representations (ICLR 2024), 2024. URL https://openreview.net/forum?id=MeB86edZIP.